# Near-Optimal Quantum Algorithms for Computing (Coarse) Correlated Equilibria of General-Sum Games

**Tongyang Li**

Center on Frontiers of Computing Studies,
and School of Computer Science,
Peking University,
Beijing, China
tongyangli@pku.edu.cn

**Xinzhao Wang**

Center on Frontiers of Computing Studies,
and School of Computer Science,
Peking University,
Beijing, China
wangxz@stu.pku.edu.cn

**Yexin Zhang**

Center on Frontiers of Computing Studies,
and School of Computer Science,
Peking University,
Beijing, China
zhangyexin@stu.pku.edu.cn

## Abstract

Computing Nash equilibria of zero-sum games in classical and quantum settings is extensively studied. For general-sum games, computing Nash equilibria is PPAD-hard and the computing of a more general concept called correlated equilibria has been widely explored in game theory. In this paper, we initiate the study of quantum algorithms for computing $\varepsilon$-approximate correlated equilibria (CE) and coarse correlated equilibria (CCE) in multi-player normal-form games. Our approach utilizes quantum improvements to the multi-scale Multiplicative Weight Update (MWU) method for CE calculations, achieving a query complexity of $\tilde{O}(m\sqrt{n})$ for fixed $\varepsilon$. For CCE, we extend techniques from quantum algorithms for zero-sum games to multi-player settings, achieving query complexity $\tilde{O}(m\sqrt{n}/\varepsilon^{2.5})$. Both algorithms demonstrate a near-optimal scaling in the number of players $m$ and actions $n$, as confirmed by our quantum query lower bounds.

## 1   Introduction

**Motivations.** Game theory is a branch of mathematics that studies the interactions between strategies of rational decision-makers. It focuses on the situations where the outcome of each participant depends on not only their own strategies but also the strategies of others. One of the simplest scenarios is a two-player zero-sum game, where the total payoff of the two players does not change regardless of their individual strategies. A key concept in game theory is *Nash equilibrium*, which describes a situation where no player can unilaterally change their strategy to achieve a better payoff, with the strategies of the other players being fixed. Notably, a Nash equilibrium in a two-player zero-sum game can be reached by no-regret online learning: when both players repeatedly adjust their strategies to minimize regret, the average play converges to the equilibrium. This observation is central to the design of several classical and quantum algorithms for computing equilibria.

39th Conference on Neural Information Processing Systems (NeurIPS 2025).

Table 1: Loss matrix where $\mathcal{D} = \{p(C, A) = \frac{1}{2}, p(B, B) = \frac{1}{2}\}$ is a CCE but not a CE: Player 1 can change $B \to D$ and $C \to A$ to reduce the loss.

|  | | Player 2 | | |
|---|---|---|---|---|
|  | A | B | C | D |
| A | (1,2) | (3,2) | (2,2) | (2,2) |
| B | (2,2) | (2,2) | (2,2) | (2,2) |
| C | (2,2) | (2,2) | (2,2) | (2,2) |
| D | (3,2) | (1,2) | (2,2) | (2,2) |

Player 1 labels rows A, B, C, D.

Grigoriadis and Khachiyan [19] showed that finding a pair of $\varepsilon$-near Nash equilibrium strategies of a two-player zero-sum game with $n$ actions could be realized using $O(n/\varepsilon^2)$ classical queries, which is sub-linear with respect to the problem size. For quantum algorithms, Refs. [25] and [3] achieved a quadratic speedup in $n$ with $\tilde{O}(\sqrt{n}/\varepsilon^4)$ and $\tilde{O}(\sqrt{n}/\varepsilon^3)$ quantum queries, respectively, and the optimality in $n$ is proven in Li et al. [25]. Currently, the state-of-the-art results [9, 18] have improved the $\varepsilon$-dependency of the query complexity to $\tilde{O}(\sqrt{n}/\varepsilon^{2.5})$.

Many scenarios in game theory cannot be modeled as two-player zero-sum games, such as the congestion game [31] and the scheduling game [16, 28]. In a congestion game, each player chooses a strategy from a set of actions, and the loss of each player depends on the number of players choosing the same action. The congestion game is a widely used model in traffic routing. In a scheduling game, strategies are a set of machines and the loss of choosing a machine depends on the total load of the machine. Both congestion games and scheduling games are examples of *normal-form games*. In an $m$-player normal-form game, player $i$ chooses a strategy $a_i$ in $\mathcal{A}_i$ with $n$ actions, and then suffers a loss $\mathcal{L}_i(a_1, \ldots, a_m)$.

For a general normal-form game, finding a Nash equilibrium is PPAD-hard [12]. A more general concept than the Nash equilibrium is the correlated equilibrium proposed by Aumann [4]. In this setting, a trusted coordinator pulls an action profile from a distribution $\mathcal{D}$ on the joint action set of all players and sends each player its action. We call $\mathcal{D}$ an $\varepsilon$-*correlated equilibrium* (CE) if no player can reduce its loss by $\varepsilon$ by changing their action based on what the coordinator sends. For any player, if it cannot reduce its loss by $\varepsilon$ by choosing a fixed action regardless of what the coordinator sends, we call the distribution $\mathcal{D}$ an $\varepsilon$-*coarse correlated equilibrium* (CCE). The coarse correlated equilibrium is a relaxation of the correlated equilibrium, hence it is easier to find one (see Table 1).

Computing the correlated equilibrium and coarse correlated equilibrium of a normal-form game has been extensively studied in the classical setting. Since the size of description of a normal-form game is exponential in $m$, any algorithm needs $\Omega(\exp(m))$ time to solve the problem in the worst case. A standard approach to handle this issue is to assume that the algorithm can query the loss function of the game as a black-box and study the query complexity of the problem. In this case, a correlated equilibrium can be computed using $\text{poly}(n, m)$ queries by LP-based algorithms [23, 28]. The algorithm proposed by Jiang and Leyton-Brown [23] can compute an exact correlated equilibrium but the degree of its query complexity is high. Analogous to the case of Nash equilibrium, an approximate correlated equilibrium can be computed using a no-swap-regret learning algorithm [15] (see its definition in Section 2). This connection has motivated a line of research focused on designing efficient no-swap-regret algorithms in normal-form games. In particular, Dagan et al. [13], Peng and Rubinstein [29] designed the first algorithms computing an $\varepsilon$-correlated equilibrium using $\tilde{O}(mn)$ queries for a fixed precision $\varepsilon$. Similarly, an $\varepsilon$-coarse correlated equilibrium can be computed by a no-external-regret learning algorithm. While recent variants of the Multiplicative Weights Update (MWU) algorithm, such as optimistic, clairvoyant, and cautious MWU [14, 30, 32, 33], achieve remarkable $\text{polylog}(T)$ regret bounds after $T$ rounds, these bounds scale polynomially with the number of players $m$. This leads to a total query complexity that is super-linear in $m$.

Our work differs from the field of *quantum games* [22, 26, 27, 36], where players play quantum strategies and quantum equilibria are considered. In contrast, we use quantum algorithms to more efficiently find classical equilibria in purely classical games.

**Contributions.** In this paper, we initiate the study of quantum algorithms for computing the CE and CCE of multi-player normal-form games, aiming for near-optimal complexity in both the num-

ber of players $m$ and actions $n$. For computing $\varepsilon$-CE, our algorithm quantizes the state-of-the-art multi-scale MWU framework [13, 29], which provides the fastest known classical convergence for a fixed $\varepsilon$. For computing $\varepsilon$-CCE, our approach is specifically designed to achieve optimal $m$ and $n$ scaling. We therefore build upon the algorithm of Grigoriadis and Khachiyan [19], whose regret bound is crucially independent of the number of players. This choice is key to designing a quantum algorithm with a query complexity that is linear in $m$, which is optimal.

We assume that a quantum computer can access the game by querying a unitary oracle $\mathcal{O}_{\mathcal{L}}$ and study the query complexity of finding an $\varepsilon$-(coarse) correlated equilibrium.

**Definition 1.** *For an $m$-player normal-form game $(\{\mathcal{A}_i\}_{i=1}^m, \{\mathcal{L}_i\}_{i=1}^m)$, a unitary oracle $\mathcal{O}_{\mathcal{L}}$ satisfying*

$$\mathcal{O}_{\mathcal{L}}|i\rangle|a_1\rangle \cdots |a_m\rangle|0\rangle = |i\rangle|a_1\rangle \cdots |a_m\rangle|\mathcal{L}_i(a_1,\ldots,a_m)\rangle \tag{1}$$

*for all $i \in [m]$ and $a_1 \in \mathcal{A}_1, \ldots, a_m \in \mathcal{A}_m$ is an oracle of the game, and the query complexity of an algorithm is the number of queries to $\mathcal{O}_{\mathcal{L}}$.*

The unitary oracle $\mathcal{O}_{\mathcal{L}}$ can be constructed efficiently if the game has a succinct representation that allows for an efficient classical algorithm to compute the loss function $\mathcal{L}_i(a_1,\ldots,a_m)$ [6]. For example, in a congestion game, a player's loss is determined by the costs of their chosen resources, where the cost of each resource depends on the total number of players who selected it. This structure allows for efficient loss calculation. Given access to $\mathcal{O}_{\mathcal{L}}$, we state the following problem of computing an $\varepsilon$-(coarse) correlated equilibrium:

**Problem 1.** *Given an $m$-player normal-form game $(\{\mathcal{A}_i\}_{i=1}^m, \{\mathcal{L}_i\}_{i=1}^m)$ with $n$ actions for each player, an error parameter $\varepsilon > 0$, and a failure probability $\alpha > 0$, prepare a quantum state*

$$|\psi_o\rangle = \sum_{a \in \mathcal{A}} \sqrt{q(a)}|a\rangle|\psi_a\rangle \tag{2}$$

*for some normalized states $|\psi_a\rangle$ such that $q$ is an $\varepsilon$-(coarse) correlated equilibrium of the game with success probability at least $1 - \alpha$.*

In particular, we give quantum algorithms for computing $\varepsilon$-CE and $\varepsilon$-CCE as follows:

**Theorem 1** (Informal version of Theorem 8)**.** *Algorithm 1 computes an $\varepsilon$-correlated equilibrium of an $m$-player normal-form game with $n$ actions for each player using $m\sqrt{n}(\log(mn))^{O(1/\varepsilon)}$ queries to $\mathcal{O}_{\mathcal{L}}$ and $m^2\sqrt{n}(\log(mn))^{O(1/\varepsilon)}$ time.*

**Theorem 2** (Informal version of Theorem 9)**.** *Algorithm 2 outputs the classical description of an $\varepsilon$-coarse correlated equilibrium of an $m$-player normal-form game with $n$ actions for each player using $\tilde{O}(m\sqrt{n}/\varepsilon^{2.5})$ queries to $\mathcal{O}_{\mathcal{L}}$ and $\tilde{O}(m^2\sqrt{n}/\varepsilon^{4.5})$ time.*

We measure the time complexity by the number of one and two-qubit gates in the quantum algorithm. The overhead in time complexity, in comparison to query complexity, arises from the gate complexity of the QRAM. If we adopt the convention established by previous quantum algorithms for zero-sum games [3, 9, 18], which assumes that QRAM access incurs a unit cost, then the time complexities presented in Theorem 1 and Theorem 2 align with the query complexities, differing only by a poly-logarithmic factor. In addition, we note that the output of Algorithm 2 is a classical description of a $\tilde{O}(B^2/\varepsilon^2)$-sparse $\varepsilon$-coarse correlated equilibrium, hence we can prepare the state $|\psi_o\rangle$ in Problem 1 in $\tilde{O}(mB^2/\varepsilon^2)$ time [20].

On the other hand, we prove the following quantum lower bounds on computing CE and CCE:

**Theorem 3** (Restatement of Theorem 7)**.** *For an $m$-player normal-form game with $n$ actions for each player, let $B$ denote an upper bound on the loss function. Assume $0 < \varepsilon < \min\{\frac{1}{3}, \frac{2B}{3m}\}$, to compute an $\varepsilon$-(coarse) correlated equilibrium with success probability more than $\frac{2}{3}$, we need $\Omega(m\sqrt{n})$ quantum queries.*

The scaling of our query complexity lower bounds with respect to the number of players $m$ and actions $n$ matches our algorithms' upper bounds up to a poly-logarithm factor, indicating the near-optimality of our quantum algorithms in $m$ and $n$.

Table 2: Complexity bounds for computing $\varepsilon$-CE.

| Reference | Setting | Query complexity | Time complexity |
|-----------|---------|------------------|-----------------|
| [13, 29] | classical | $mn(\log(mn))^{O(1/\varepsilon)}$ | $mn(\log(mn))^{O(1/\varepsilon)}$ |
| this paper | quantum | $m\sqrt{n}(\log(mn))^{O(1/\varepsilon)}, \Omega(m\sqrt{n})$ | $m^2\sqrt{n}(\log(mn))^{O(1/\varepsilon)}$ |

Table 3: Complexity bounds for computing $\varepsilon$-CCE.

| Reference | Setting | Query complexity | Time complexity |
|-----------|---------|------------------|-----------------|
| [19][1] | classical | $\tilde{O}(mn/\varepsilon^2)$ | $\tilde{O}(mn/\varepsilon^2)$ |
| this paper | quantum | $\tilde{O}(m\sqrt{n}/\varepsilon^{2.5}), \Omega(m\sqrt{n})$ | $\tilde{O}(m^2\sqrt{n}/\varepsilon^{4.5})$ |

**Techniques.** Our algorithm for CE quantizes the multi-scale MWU algorithm [29]. Classical multi-scale MWU algorithm needs $\Omega(n)$ queries to compute the loss vector of one player in each round and then takes the exponential of the loss vector to update its strategy. This $\Omega(n)$ query complexity can be improved in quantum algorithms by constructing an amplitude-encoding of the loss vector and then using the quantum Gibbs sampler to sample from the exponential of the loss vector. The standard approach to construct the amplitude-encoding is to store the frequency of history action samples in a QRAM and maintain a tree data structure [3, 9, 18]. However, in an $m$-player normal-form game with $n$ actions for each player, the size of the joint action space is $n^m$, so the QRAM requires $\Omega(n^m)$ gates to implement. Furthermore, the multi-scale MWU algorithm runs $O(1/\varepsilon)$ instances of the MWU algorithm in parallel, thus standard amplitude-encoding schemes require $O(1/\varepsilon)$ QRAMs to store the frequency of history action samples in different time intervals for different MWU instances. To overcome these issues, we use a single, unified QRAM to store all history action samples rather than the frequency vector. We then demonstrate how the necessary amplitude-encoding for any MWU subroutine can be constructed from this single QRAM. Crucially, instead of treating QRAM access as a unit-cost oracle, we analyze its gate-level construction cost, showing that it requires only $m \log n(\log(mn))^{O(1/\varepsilon)}$ gates.

Our algorithm for CCE is built upon the quantum algorithm by Bouland et al. [9], which quantizes the classical approach of Grigoriadis and Khachiyan [19] for two-player zero-sum games. We extend their quantum framework to the $m$-player normal-form game setting, using the "ghost iteration" technique to prove that the algorithm converges to an $\varepsilon$-CCE in $\tilde{O}(1/\varepsilon^2)$ iterations. We adapt the amplitude-encoding schemes from our CE algorithm to avoid the exponential gate overhead in the QRAM construction.

For the lower bound, we reduce the direct product of $m$ instances of the unstructured search problem to the problem of computing an $\varepsilon$-CE ($\varepsilon$-CCE) of an $m$-player normal-form game. Then, we combine the lower bound on the unstructured search problem [7] with the direct product theorem [24] to prove the lower bound on computing an $\varepsilon$-CE ($\varepsilon$-CCE) of an $m$-player normal-form game.

**Open questions.** Our results leave several natural open questions for future investigation:

- An open question is whether the $\varepsilon$ dependence of our CCE algorithm can be improved. While quantizing the optimistic MWU algorithm of Daskalakis et al. [14] is a natural target, its analysis relies on high-order smoothness properties of the loss vectors. These properties are highly sensitive to the sampling noise introduced by a quantum Gibbs sampler, making a direct quantization challenging (see the discussion in Appendix D). A more promising direction would be to quantize the Regularized Value Update (RVU) framework of Syrgkanis et al. [35], which relies on more robust first-order properties. This could serve as a crucial first step towards quantizing recent, highly-efficient algorithms like Cautious MWU [32], which build upon the RVU framework.

- Beyond normal-form games, equilibria of Bayesian games and extensive-form games are also studied in game theory [17, 37]. Dagan et al. [13], Peng and Rubinstein [29] showed

---

[1]This algorithm is designed for computing $\varepsilon$-Nash equilibrium of a two-player zero-sum game, but we show in Corollary 1 that it can be used to compute an $\varepsilon$-CCE of a multi-player normal-form game.

that an $\varepsilon$-CE in extensive-form games can be computed efficiently. Can we design quantum algorithms to compute the equilibrium of Bayesian games and extensive-form games with quantum speedup?

- Can we reduce the time complexity of computing $\varepsilon$-CE and $\varepsilon$-CCE to $\tilde{O}(m\sqrt{n})$ which aligns with our query complexity? The difficulty is that we need to sample strategies for all $m$ players in each round of the game, and each call to the quantum Gibbs sampler requires access to the QRAM, incurring an overhead of $O(m)$.

## 2 Preliminaries

### 2.1 Game theory and no-regret learning

**Game theory** An $m$-player normal-form game can be described by a tuple $(\{\mathcal{A}_i\}_{i=1}^m, \{\mathcal{L}_i\}_{i=1}^m)$, where $\mathcal{A}_i$ with $|\mathcal{A}_i| = n$ is the action set of player $i$ and $\mathcal{L}_i$ is the loss function of player $i$. Without loss of generality, we let $\mathcal{A}_i = [n]$. Let $\mathcal{A} = \mathcal{A}_1 \times \cdots \times \mathcal{A}_m$ be the joint action set. The loss function of player $i$ is a function $\mathcal{L}_i \colon \mathcal{A} \to [0, B]$, representing the loss of player $i$; here $B$ is an upper bound on loss functions. For an action profile $a = (a_1, \ldots, a_m) \in \mathcal{A}$, let $a_{-i}$ denote the profile after removing $a_i$. For any finite set $S$, we let $\Delta(S)$ denote the probability simplex over $S$. In each round of the game, player $i$ can choose an independent mixed strategy $x_i \in \Delta(\mathcal{A}_i)$. The collection of these strategies, $x = (x_1, \ldots, x_m)$, is called a mixed strategy profile and induces a product distribution over $\mathcal{A}$. A more general concept is a correlated strategy, which is any joint distribution $\mathcal{D} \in \Delta(\mathcal{A})$. For a mixed strategy profile $x$, we let $x_{-i}$ denote the profile after removing $x_i$.

We consider two types of equilibria in normal-form games: correlated equilibrium and coarse correlated equilibrium.

**Definition 2** (Correlated equilibrium). *For an $m$-player normal-form game $(\{\mathcal{A}_i\}_{i=1}^m, \{\mathcal{L}_i\}_{i=1}^m)$, a distribution $\mathcal{D} \in \Delta(\mathcal{A})$ is called an $\varepsilon$-correlated equilibrium if for any $i \in [m]$ and any function $\phi_i \colon \mathcal{A}_i \to \mathcal{A}_i$*

$$\mathop{\mathbb{E}}_{a\sim\mathcal{D}}[\mathcal{L}_i(a_i, a_{-i})] \leq \mathop{\mathbb{E}}_{a\sim\mathcal{D}}[\mathcal{L}_i(\phi_i(a_i), a_{-i})] + \varepsilon. \tag{3}$$

**Definition 3** (Coarse correlated equilibrium). *For an $m$-player normal-form game $(\{\mathcal{A}_i\}_{i=1}^m, \{\mathcal{L}_i\}_{i=1}^m)$, a distribution $\mathcal{D} \in \Delta(\mathcal{A})$ is called an $\varepsilon$-coarse correlated equilibrium if for any $i \in [m]$ and $a_i' \in \mathcal{A}_i$*

$$\mathop{\mathbb{E}}_{a\sim\mathcal{D}}[\mathcal{L}_i(a_i, a_{-i})] \leq \mathop{\mathbb{E}}_{a\sim\mathcal{D}}[\mathcal{L}_i(a_i', a_{-i})] + \varepsilon. \tag{4}$$

**Online learning** In the adversarial online learning setting, a player plays against an adversary sequentially for $T$ rounds. In the $t$-th round, the player plays a distribution $x^{(t)}$ over its action set $[n]$. Then, the adversary selects a loss vector $\ell^{(t)} \in [0, B]^n$ and the player suffers from a loss $\langle x^{(t)}, \ell^{(t)} \rangle$. The player observes the loss vector $\ell^{(t)}$ and updates its strategy based on the previous loss vectors to minimize its total regret in $T$ rounds. We consider two kinds of regret: the standard *external regret* and the *swap regret*. The external regret of player $i$ is defined as

$$\text{Regret}_{i,T} = \sum_{t=1}^T \langle x_i^{(t)}, \ell_i^{(t)} \rangle - \min_{x_i \in \Delta(\mathcal{A}_i)} \sum_{t=1}^T \langle x_i, \ell_i^{(t)} \rangle, \tag{5}$$

which measures the maximum reduction in loss that could be achieved by switching to a fixed action strategy. Let $\Phi_i$ denote the set of functions $\phi \colon [n] \to [n]$. The swap regret of player $i$ is defined as

$$\text{Swap-Regret}_{i,T} = \sum_{t=1}^T \langle x_i^{(t)}, \ell_i^{(t)} \rangle - \min_{\phi \in \Phi_i} \sum_{t=1}^T \sum_{j=1}^n x_i^{(t)}(j) \cdot \ell_i^{(t)}(\phi(j)), \tag{6}$$

which measures the maximum reduction in loss that could be achieved by using a fixed swap function on its history strategies. An algorithm is called a *no-regret* learning algorithm if the total regret is $o(T)$. The Multiplicative Weight Update (MWU) algorithm is a well-known no-external-regret learning algorithm. It updates the strategy by multiplying the previous strategy by the exponential of the negative sum of the loss vectors. This ensures that actions with lower cumulative loss are favored over time, achieving $O(\sqrt{T \log n})$ external regret. The detailed procedure is shown in Appendix A as Algorithm 3.

**Theorem 4** (Theorem 1.5 in [21]). *The external regret of the MWU algorithm (Algorithm 3) with step size $\eta = \sqrt{\log n / T}/B$ is at most $2B\sqrt{T \log n}$.*

The multi-scale MWU algorithm, proposed by Peng and Rubinstein [29], achieves $\mathrm{polylog}(n)$ swap regret by running multiple instances of the MWU algorithm in parallel at different time scales. Each instance aggregates losses over increasingly longer intervals before performing an update, and the final strategy is a uniform mixture of the strategies from each instance. The detailed procedure is shown in Appendix A as Algorithm 4.

**Theorem 5** (Theorem 1.1 in [29]). *For any $\varepsilon > 0$, the multi-scale MWU algorithm (Algorithm 4) has at most $\varepsilon BT$ swap regret in $T = (16 \log(n)/\varepsilon^2)^{2/\varepsilon}$ rounds.*

**No-regret learning in normal-form games** It is known that if all players play according to a no-regret learning algorithm with external (or swap) regret at most $\varepsilon(T)$ in $T$ rounds, then the uniform mixture of their strategies in all $T$ rounds is an $O(\varepsilon(T)/T)$-approximation of a coarse correlated equilibrium (or correlated equilibrium) of the game (see Section 7 of [10]).

For the external regret, when loss vectors are adversarial, the celebrated MWU algorithm guarantees $O(\sqrt{T \log n})$ regret, which is optimal (see Section 3.7 of [10]). However, in the setting of repeated game playing, algorithms with recency bias can do better due to the smoothness of the loss vectors. Syrgkanis et al. [35] showed that if all $m$ players run an algorithm from a specific class of algorithms with recency bias, then each player experiences $O(\log n \cdot \sqrt{m} \cdot T^{1/4})$ external regret. Chen and Peng [11] improved this bound to $O(\log^{5/6} n \cdot T^{1/6})$ in two-player normal-form games when both players run the optimistic MWU algorithm. Daskalakis et al. [14] then dramatically improved the $T$ dependency by showing that if all players run the optimistic MWU algorithm in an $m$-player normal-form game, each player experiences $O(\log n \cdot m \cdot \log^4 T)$ external regret, so the uniform mixture of their strategies is a $\tilde{O}(m \log n/T)$-coarse correlated equilibrium after $T$ rounds. The dependence on $T$ is further improved by subsequent algorithms like Clairvoyant, and Cautious MWU [30, 32, 33].

It is known that an external-regret minimization algorithm can be converted to a swap-regret minimization algorithm [8, 34]. Chen and Peng [11], Anagnostides et al. [1], and Anagnostides et al. [2] used this reduction to design algorithms with $O(T^{1/4})$, $O(\log^4 T)$, and $O(\log T)$ swap-regret respectively in an $m$-player normal-form game if other players run the same algorithm. However, this reduction incurs an $\Omega(n)$ overhead. Dagan et al. [13] improved this reduction and proposed an algorithm that has at most $\varepsilon T$ swap regret in $T = (\log n/\varepsilon^2)^{O(1/\varepsilon)}$ rounds in the standard adversarial online learning setting, which aligns with the upper bound in Peng and Rubinstein [29].

## 2.2 Quantum computing

The fundamental unit of information in quantum computing is the quantum bit or qubit. Unlike classical bits that are either 0 or 1, a qubit can exist in a superposition of states, represented as a unit vector in a two-dimensional complex Hilbert space: $|\psi\rangle = \alpha|0\rangle + \beta|1\rangle$, where $\{|0\rangle, |1\rangle\}$ forms a (orthonormal) computational basis, and the amplitudes $\alpha, \beta \in \mathbb{C}$ satisfy $|\alpha|^2 + |\beta|^2 = 1$. An $n$-qubit quantum system resides in the tensor product space of $n$ Hilbert space $\mathbb{C}^2$, which can be written as $(\mathbb{C}^2)^{\otimes n} = \mathbb{C}^{2^n}$ with computational basis states $\{|i\rangle\}_{i=0}^{2^n-1}$, and a quantum state of $n$ qubits can therefore represent a superposition of all $2^n$ possible states: $|\psi\rangle = \sum_{i=0}^{2^n-1} \alpha_i|i\rangle$, where $\sum_i |\alpha_i|^2 = 1$. Information can be obtained by quantum measurement on a computational basis, where measuring state $|\psi\rangle = \sum_{i=0}^{2^n-1} \alpha_i|i\rangle$ on basis $\{|i\rangle\}$ yields outcome $i$ with probability $p(i) = |\alpha_i|^2$ for every $i \in [2^n]$. Quantum states evolve through unitary transformations: $|\psi\rangle \to U|\psi\rangle$, where $U \in \mathbb{C}^{2^n \times 2^n}$ is a unitary satisfying $UU^\dagger = U^\dagger U = I_{2^n}$, where $U^\dagger$ is the Hermitian conjugate of operator $U$. For two quantum states $|\psi\rangle = \sum_{i=0}^{2^n-1} \alpha_i|i\rangle$ and $|\phi\rangle = \sum_{i=0}^{2^n-1} \beta_i|i\rangle$, their inner product is defined by $\langle i|\psi\rangle = \sum_i \alpha_i^* \beta_i$. The tensor product of two quantum states $|\psi\rangle \in \mathbb{C}^{d_1}$ and $|\phi\rangle \in \mathbb{C}^{d_2}$ is denoted as $|\psi\rangle|\phi\rangle = |\psi\rangle \otimes |\phi\rangle \in \mathbb{C}^{d_1 d_2}$.

In the quantum query model, an algorithm accesses the given function $f$ via a quantum oracle. This oracle, denoted $\mathcal{O}_f$, is defined as a unitary operator that performs the following reversible computation on the computational basis states: $\mathcal{O}_f|x\rangle|0\rangle = |x\rangle|f(x)\rangle$. A key advantage of this model is that the oracle can be queried on a superposition of inputs.

The term QRAM can refer to several distinct models in quantum computing. In this work, we use "QRAM" to refer specifically to a circuit providing quantum access to classical data, a model more precisely known as Quantum Read-Only Memory (QROM). We retain the more common term

QRAM and the notation $U_{\text{QRAM}}$ throughout this paper for consistency with related literature. Formally, for a memory containing $N$ classical bitstrings $\{D_i\}_{i=0}^{N-1}$, this unitary performs the mapping $U_{\text{QRAM}}|i\rangle|0\rangle \mapsto |i\rangle|D_i\rangle$. Such circuits can be constructed from elementary gates with a complexity linear in $N$ and the bit-length of the entries [5].

### 2.3 Quantum algorithms for games

Quantum algorithms for finding Nash equilibria in zero-sum games have been well-studied [3, 9, 18], and achieve a quadratic speedup in $n$. Most of the quantum algorithms for games quantize variants of the MWU algorithm. Note that the strategies output by the MWU algorithm can be written as an exponential of the accumulated loss vectors. For a classical MWU algorithm, computing the exponential of a vector $u \in \mathbb{R}^n$ requires $\Omega(n)$ time. To reduce this overhead, quantum algorithms use the quantum Gibbs sampler. Suppose that the quantum algorithm can access a unitary operator $V$ which encodes the vector $u$:

**Definition 4** (Amplitude encoding). *A unitary operator $V$ is said to be a $\beta$-amplitude-encoding of a vector $u \in \mathbb{R}^n$ with non-negative entries, if*

$$\langle 0|_C V|0\rangle_C|i\rangle_A|0\rangle_B = \sqrt{\tfrac{u_i}{\beta}}|i\rangle_A|\psi_i\rangle_B \tag{7}$$

*for all $i \in [n]$. Here, $|\psi_i\rangle_B$ is a normalized garbage state in an ancilla register.*

Then the quantum Gibbs sampler can prepare the state $\sum_{i=1}^n \sqrt{q_i}|i\rangle|\psi_i\rangle$ where the distribution $q = (q_1, \ldots, q_n)$ is close to $\exp(-u)/\|\exp(-u)\|_1$, and measuring the first register gives a sample approximately from the distribution $\exp(-u)/\|\exp(-u)\|_1$.

**Theorem 6** (Quantum Gibbs sampler [18]). *Given access to a unitary $V$ which is a $\beta$-amplitude-encoding of a vector $u \in \mathbb{R}^n$, there is a unitary oracle $\mathcal{O}_u^{\text{Gibbs}}(\delta)$ such that*

$$\mathcal{O}_u^{\text{Gibbs}}(\delta) \colon |0\rangle|0\rangle \mapsto \sum_{i=1}^n \sqrt{q_i}|i\rangle|\psi_i\rangle \tag{8}$$

*where $q$ is $\delta$-close to $\exp(-u)/\|\exp(-u)\|_1$ in total variation distance. $\mathcal{O}_u^{\text{Gibbs}}(\delta)$ can be implemented using $\tilde{O}(\beta\sqrt{n})$ queries to $V$ and $\tilde{O}(\beta\sqrt{n})$ time.*

In many game-solving algorithms that use Gibbs sampling, the underlying vector $u$ changes slowly, often receiving only sparse updates in each round. Based on this property, Bouland et al. [9] proposed a dynamic Gibbs sampler, an oracle $\mathcal{O}_u^{\text{dynamic-Gibbs}}$ for repeatedly sampling from a distribution that is $\delta$-close to the changing Gibbs distribution $\exp(u)/\|\exp(u)\|_1$.

**Problem 2** (Sampling maintenance for two-player game, Problem 1 in [9]). *Given $\eta > 0, 0 < \delta < 1$, and access to a quantum oracle for $A \in \mathbb{R}^{n_1 \times n_2}$. Now consider a sequence of size $T$, where each item includes an "Update" operation to a dynamic vector $x \in \mathbb{R}_{\geq 0}^{n_2}$, each in the form of $x_i \leftarrow x_i + \eta$ for some $i \in [n_2]$. The goal is to maintain a $\delta$-approximate Gibbs oracle $\mathcal{O}_{Ax}^{\text{dynamic-Gibbs}}$ during the "Update" operations. Let $\mathcal{T}_{\text{update}}$ denote queries per operation we need, and let $\mathcal{T}_{\text{samp}}$ denote the worst-case time needed for $\mathcal{O}_{Ax}^{\text{dynamic-Gibbs}}$.*

Bouland et al. [9] provided an efficient solution to Problem 2 using a special data structure to store partial information of the Gibbs distribution and maintaining its effectiveness across many rounds to reduce the amortized complexity of each sampling.

## 3 Quantum algorithm for computing correlated equilibria

In this section, we present the quantum algorithm (Algorithm 1) for computing an $\varepsilon$-correlated equilibrium ($\varepsilon$-CE) in a normal-form game. The high-level ideas of the algorithm are presented below. The implementation details, as well as the formal proof of the algorithms correctness and complexity analysis, are provided in Appendix B.1 and Appendix B.2.

Our quantum algorithm (Algorithm 1) for computing an $\varepsilon$-CE of a normal form game improves on the protocol in Peng and Rubinstein [29]. The algorithm simulates $m$ players playing the game repeatedly and using the multi-scale MWU (Algorithm 4) algorithm to update their strategies. At the $t$-th round, to estimate the loss vectors, we use quantum Gibbs sampler to sample from the joint

distribution of all players' strategies at this round. We sample $S$ action profiles $a^{(t,1)}, \ldots, a^{(t,S)} \in \mathcal{A}$ and let the loss vector of player $i$ at the $t$-th round be

$$\ell_{i,t}(a_i) := \tfrac{1}{S} \sum_{s=1}^{S} \mathcal{L}_i(a_i; a_{-i}^{(t,s)}) \quad \forall a_i \in \mathcal{A}_i. \tag{9}$$

Let $K = \lceil \log_2(1/\varepsilon) + 1 \rceil$, $H = \lceil 4 \log(n) 2^{2K} \rceil$ be the internal parameters. According to the update rule of the multi-scale MWU, the strategy $p_{i,t}$ of player $i$ is determined by its accumulated loss vectors:

$$p_{i,t} := \tfrac{1}{2^K} \sum_{k \in [2^K]} q_{i,k,t}, \tag{10}$$

where

$$q_{i,k,t}(a_i) \propto \exp\big(-\sqrt{\log n/H} \sum_{t=(r_{k,t}-1)H^k+1}^{(r_{k,t}-1)H^k+h_{k,t}H^{k-1}} \ell_{i,t}(a_i)\big) \quad \forall a_i \in \mathcal{A}_i, \tag{11}$$

$r_{k,t}$ and $h_{k,t}$ correspond to the parameters $r$ and $h$ in subroutine $\mathrm{MWU}_k$ in the $t$-th round of Algorithm 4, and they can be computed from $t$ and $k$. The complexity bottleneck of the classical protocol in [29] lies in computing the $n$-dimensional vector $q_{i,k,t}$, which requires $\Omega(n)$ queries. To achieve sublinear quantum query complexity in $n$, we store the historical samples in a quantum random access memory (QRAM) and use a quantum Gibbs sampler to approximately sample from the distribution $p_{i,t}$.

---

**Algorithm 1** Sample-based multi-scale MWU for CE

1: **Input parameters** $m$ (number of players), $n$ (number of actions), $\varepsilon$ (error parameter), $B$ (bound on loss functions), $\alpha$ (failure probability)
2: **Internal parameters** $K := \lceil \log_2(3B/\varepsilon) + 1 \rceil$, $H := \lceil 4 \log(n) 2^{2K} \rceil$, $T := H^{2^K}$, $S := \left\lceil \frac{18B^2}{\varepsilon^2} \log\left(\frac{2mnT}{\alpha}\right) \right\rceil$, $\delta := \varepsilon/6B$
3: **Output** quantum state $|\psi_o\rangle$
4: **for** $t = 1, \ldots, T$ **do**
5:     Obtain a unitary $V_t$ such that for any $i \in [m]$ and $k \in [2^K]$, $\langle k|\langle i|V_t|k\rangle|i\rangle$ is a $(Bh_{k,t}H^{k-1})$-amplitude-encoding of the vector

$$\bar{\ell}_{k,t} := \left( \sum_{\tau=(r_{k,t}-1)H^k+1}^{(r_{k,t}-1)H^k+h_{k,t}H^{k-1}} \tfrac{1}{S} \sum_{s=1}^{S} \mathcal{L}_i(a_i; a_{-i}^{(\tau,s)}) \right)_{a_i \in \mathcal{A}_i} \tag{12}$$

    such that

$$r_{k,t} = \left\lceil \tfrac{t}{H^k} \right\rceil, \quad h_{k,t} = \left\lfloor \tfrac{t-(r_{k,t}-1)H^k}{H^{k-1}} \right\rfloor. \tag{13}$$

6:     For any $i \in [m]$, independently obtain $S$ samples $a_i^{(t,1)}, \ldots, a_i^{(t,S)}$ from the Gibbs sampling oracle $\mathcal{O}^{\mathrm{Gibbs}}_{\sqrt{\log n/H}\bar{\ell}_{k,t}}(\delta)$ with uniformly random $k \in [2^K]$.
7:     Store the samples $a^{(t,s)} = (a_i^{(t,s)})_{i \in [m]}$ for $s \in [S]$ in the QRAM.
8: **end for**
9: Prepare the uniform superposition of $t \in [T]$ and $k \in [2^K]$:

$$\frac{1}{\sqrt{T2^K}} \sum_{t=1}^{T} \sum_{k=1}^{2^K} |t\rangle|k\rangle \bigotimes_{i \in [m]} |0\rangle_{A_i} |0\rangle_{B_i}. \tag{14}$$

    Apply $\mathcal{O}^{\mathrm{Gibbs}}_{\sqrt{\log n/H}\bar{\ell}_{k,t}}(\delta)$ to register $A_i$ and $B_i$ for all $i \in [m]$. Denote $|\psi_o\rangle$ as the resulting state.
10: **return** the state $|\psi_o\rangle$.

---

## 4 Quantum algorithm for computing coarse correlated equilibria

In this section, we present the quantum algorithm (Algorithm 2) for computing an $\varepsilon$-coarse correlated equilibrium ($\varepsilon$-CCE) in a normal-form game. The high-level ideas of the algorithm are outlined

below. The algorithmic details and the formal proof of correctness and complexity are provided in Appendix B.3 and Appendix B.4.

Our quantum algorithm (Algorithm 2) for computing an $\varepsilon$-coarse correlated equilibrium of a normal-form game improves on the classic algorithm in Grigoriadis and Khachiyan [19] using approximate quantum Gibbs sampling instead of exact computation. The main technique we use is stochastic mirror descent, into which we incorporate an approximate Gibbs sampling. In each round of the algorithm, we perform a Gibbs sampling on the current weight of each player, using the sampling result to minimize the first-order approximation of loss function with the added KL divergence term at current strategy for each player. At a high level, for the strategy $u_i^{(t)}$ obtained in each round, our update method satisfies

$$u_i^{(t+1)} \approx \arg\min_{u_i} \left\{ \langle \mathcal{L}_i(j, u_{-i}^{(t)}), u_i \rangle + \sum_{j \in [n]} [u_i]_j \log \frac{[u_i]_j}{[u_i^{(t)}]_j} \right\}. \tag{15}$$

In the two-player game setting considered in Bouland et al. [9], a sampler tree data structure is employed, where for each player, an $n$-dimensional vector is maintained to record the opponents strategies over previous rounds. Extending this to an $m$-player game presents a significant challenge, since the number of opponent strategies is on the order of $n^{m-1}$. To enable an efficient dynamic Gibbs sampler, we improve upon this approach by leveraging QRAM to directly store the strategies from each round. This allows us to achieve the same functionality as the sampler tree with identical query complexity but improved time complexity. See Appendix B.3 for our implementation.

---

**Algorithm 2** Sample-based MWU for CCE

---

1: **Input parameters** $m$ (number of players), $n$ (number of actions), $\varepsilon$ (error parameter), $\alpha$ (failure probability)
2: **Internal parameters** $T := \left\lceil \max\left\{ \frac{64B^2 \log n}{\varepsilon^2}, \frac{512B^2 \log(4/\alpha)}{\varepsilon^2} \right\} \right\rceil$, $\eta := \sqrt{\log n / T}/B$, $\delta := \frac{\varepsilon}{16B(n-1)}$
3: **Output** $(\hat{x}_i)_{i \in [m]}$
4: Initialize $\hat{x}_i \leftarrow \mathbf{0}_n$.
5: **for** $t = 0, \ldots, T-1$ **do**
6:    Independently sample $a_i^{(t)}$ from $\mathcal{O}_{-\eta \cdot \sum_{k=0}^{t-1} \mathcal{L}(j, a_{-i}^{(k)})}^{\text{dynamic-Gibbs}}(\delta)$ for $i \in [m]$ and set $a^{(t)} = (a_1^{(t)}, \ldots, a_m^{(t)})$.
7:    Store the sample $a^{(t)}$ in the QRAM.
8:    Update $\hat{x}_i = \hat{x}_i + e_{a_i^{(t)}}/T$ for $i \in [m]$.
9: **end for**
10: **return** $(\hat{x}_i)_{i \in [m]}$.

---

Note that if using exact oracles of Gibbs sampling, the main skeleton of the algorithm is a natural extension of Grigoriadis and Khachiyan [19] from two-player games to multi-player games.

**Corollary 1.** *There exists a classical algorithm that computes an $\varepsilon$-coarse correlated equilibrium with high probability using $\tilde{O}(mn/\varepsilon^2)$ classical queries to $\mathcal{L}$.*

## 5 Quantum lower bounds

In this section, we prove quantum query lower bounds on finding correlated equilibria and coarse correlated equilibria.

### 5.1 Quantum lower bound for computing correlated equilibria

For computing correlated equilibria, Algorithm 4 solves the problem using $\tilde{O}(m\sqrt{n})$ queries. To complement this upper bound, we prove a matching quantum lower bound (up to poly-logarithmic factors) in $m$ and $n$.

**Theorem 7.** *Let $B$ denote the bound of loss functions. Assume $0 < \varepsilon < \min\{\frac{1}{3}, \frac{2B}{3m}\}$. For an $m$-player normal-form game with $n$ actions for each player, to return an $\varepsilon$-correlated equilibrium with success probability more than $\frac{2}{3}$, we need $\Omega(m\sqrt{n})$ queries to $\mathcal{O}_u$.*

To prove Theorem 7, we construct a hard instance and claim that finding an $\varepsilon$-correlated equilibrium on this instance is sufficiently difficult.

**Definition 5** (Hard Instance). *Consider an $m$-player normal-form game with $n$ actions $\{1, 2, \ldots, n\}$ for each player. Each player $i \in [m]$ selects $k_i \in [n]$ uniformly randomly, and then define the loss function as follows:*

$$\mathcal{L}_i(a_1, a_2, \ldots, a_m) = \begin{cases} 0 & \text{if } a_i = k_i; \\ B & \text{if } a_i \neq k_i. \end{cases}$$

*Here $a_i$ is the action taken by player $i$.*

The $\varepsilon$-correlated equilibrium of Definition 5 is straightforward: each player $i \in [m]$ takes action $k_i$ with probability more than $1 - \varepsilon/B$. Intuitively, each player's utility depends only on their own actions and is independent of the strategies of other players. Therefore, the goal of finding the $\varepsilon$-correlated equilibrium is essentially to determine the value of $k_i$ for each $i \in [m]$. This is similar to computing $m$ copies of a search problem on the entries of $A$ with $|A| = n$, and we will establish a query lower bound of correlated equilibria by constructing a reduction between the two problems.

**Lemma 1.** *Given an algorithm $\mathcal{A}$ finding an $\varepsilon$-correlated equilibrium of Definition 5 with success probability more than $1 - \delta$, we can solve the $m$ copies of $n$-item search problem with success probability $1 - (\delta + \frac{\varepsilon m}{B})$ applying $\mathcal{A}$ once.*

Finally, for the $m$ copies problem, Lee and Roland [24] proposed the strong direct product theorem that establishes a lower bound on the query complexity for such problems. In our setting of the correlated equilibrium problem, this lower bound corresponds to $m\sqrt{n}$, which matches the complexity of Algorithm 4. This indicates that our quantum algorithm is optimal in terms of both $m$ and $n$. The formal proof of Theorem 7 and Lemma 1 are provided in Appendix C.

### 5.2 Quantum lower bound for computing coarse correlated equilibria

Now, we consider the quantum query lower bound on finding coarse correlated equilibria. Notice that for our hard instance Definition 5, $\varepsilon$-correlated equilibria and $\varepsilon$-coarse correlated equilibria are equivalent. Therefore, we can directly derive the quantum query lower bound for finding coarse correlated equilibria from the above analysis:

**Corollary 2.** *Let $B$ denote the bound of the loss function. Assume $0 < \varepsilon < \min\{\frac{1}{3}, \frac{2B}{3m}\}$, for an $m$-player normal-form game with $n$ actions for each player, to return an $\varepsilon$-coarse correlated equilibrium with success probability more than $\frac{2}{3}$, we need $\Omega(m\sqrt{n})$ queries to $\mathcal{O}_u$.*

## Acknowledgments

This work was supported by the National Natural Science Foundation of China (Grant Number 62372006).

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

# A Omitted Algorithm Details

Below we provide the full pseudocode for the Multiplicative Weight Update (MWU) and multi-scale MWU algorithms, which are referenced in Section 2.

---

**Algorithm 3** Multiplicative Weight Update (MWU)

---

1: **Input parameters** $T$ (number of rounds), $n$ (number of actions), $B$ (bound on loss vector)
2: **for** $t = 1, \ldots, T$ **do**
3:     Set $x^{(t)} \in \Delta([n])$ such that $x^{(t)}(i) \propto \exp(-\eta \sum_{\tau=1}^{t-1} \ell^{(\tau)}(i))$ for $i \in [n]$, where $\eta = \sqrt{\log n / T}$
4:     Play $x^{(t)}$ and observe $\ell^{(t)} \in [0, B]^n$
5: **end for**

---

---

**Algorithm 4** Multi-scale MWU

---

**Input parameters** $\varepsilon$ (precision), $n$ (number of actions), $B$ (bound on the loss vector)
**Internal parameters** $K := \lceil \log_2(1/\varepsilon) + 1 \rceil$, $H := \lceil 4 \log(n) 2^{2K} \rceil$, number of rounds $T := H^{2^K}$
**for** $t = 1, \ldots, T$ **do**
    Let $q_{k,t} \in \Delta_n$ be the strategy of $\text{MWU}_k$ ($k \in [2^K]$), play uniformly over them

$$p_t = \tfrac{1}{2^K} \sum_{k \in [2^K]} q_{k,t} \tag{16}$$

**end for**
**procedure** $\text{MWU}_k$
    **for** $r = 1, 2, \ldots, T/H^k$ **do**
        Initiate MWU with input parameters $H, n, H^{k-1}B$
        **for** $h = 1, 2, \ldots, H$ **do**
            Let $z_{r,h} \in \Delta_n$ be the strategy of MWU at the $h$-th round
            Play $z_{r,h}$ for $H^{k-1}$ days
            Update MWU with the aggregated loss of the last $H^{k-1}$ days

$$\left\{ \sum_{\tau=(r-1)H^k+(h-1)H^{k-1}+1}^{(r-1)H^k+hH^{k-1}} \ell_\tau(i) \right\}_{i \in [n]} \in [0, H^{k-1}B]^n \tag{17}$$

        **end for**
    **end for**
**end procedure**

---

# B Technical details of algorithms

In this appendix, we present the implementation details and formal proofs of our main technical results, Theorem 1 and Theorem 2. Specifically, we provide the correctness and complexity analysis of our quantum algorithms, Algorithm 1 and Algorithm 2, for computing an $\varepsilon$-correlated equilibrium and an $\varepsilon$-coarse correlated equilibrium, respectively.

## B.1 Implementation of Algorithm 1

We now describe the details of the implementation of Algorithm 1. At the $t$-th round, suppose samples before the $t$-th round are stored in the QRAM. Then we can access the samples in superposition by applying the unitary $U_{\text{QRAM}}$ such that

$$U_{\text{QRAM}} \colon |\tau\rangle|s\rangle|0\rangle \mapsto |\tau\rangle|s\rangle|a^{(\tau,s)}\rangle \tag{18}$$

for any $\tau < t$ and $s \in [S]$. Given access to the QRAM, we now show how to implement the unitary $V_t$ in Algorithm 1. Since $r_{t,k}$ and $h_{t,k}$ can be computed efficiently, we can prepare the following

uniform superposition state given $k$ and $t$:

$$\frac{1}{\sqrt{h_{k,t}H^{k-1}S}} \sum_{\tau=(r_{k,t}-1)H^k+1}^{(r_{k,t}-1)H^k+h_{k,t}H^{k-1}} \sum_{s=1}^{S} |\tau\rangle|s\rangle|0\rangle. \tag{19}$$

Then applying $U_{\text{QRAM}}$, we get the uniform superposition of samples:

$$\frac{1}{\sqrt{h_{k,t}H^{k-1}S}} \sum_{\tau=(r_{k,t}-1)H^k+1}^{(r_{k,t}-1)H^k+h_{k,t}H^{k-1}} \sum_{s=1}^{S} |\tau\rangle|s\rangle|a^{(\tau,s)}\rangle. \tag{20}$$

Using one query to $\mathcal{O}_\mathcal{L}$ and $\mathcal{O}_\mathcal{L}^\dagger$, we can map

$$|i\rangle|a_i\rangle|a_{-i}\rangle|0\rangle \mapsto |i\rangle|a_i\rangle|a_{-i}\rangle\left(\sqrt{\frac{\mathcal{L}_i(a_i,a_{-i})}{B}}|1\rangle + \sqrt{1-\frac{\mathcal{L}_i(a_i,a_{-i})}{B}}|0\rangle\right). \tag{21}$$

Combining Eq. (19), Eq. (20), and Eq. (21), for any $i \in [m]$, $k \in [2^K]$ and $a_i \in \mathcal{A}_i$, we can perform the following unitary transformation:

$$V_t: |k\rangle|i\rangle|a_i\rangle|0\rangle|0\rangle|0\rangle|0\rangle$$

$$\mapsto |k\rangle|i\rangle|a_i\rangle \frac{1}{\sqrt{h_{k,t}H^{k-1}S}} \sum_{\tau=(r_{k,t}-1)H^k+1}^{(r_{k,t}-1)H^k+h_{k,t}H^{k-1}} \sum_{s=1}^{S} |\tau\rangle|s\rangle|a^{(\tau,s)}\rangle\left(\sqrt{\frac{\mathcal{L}_i(a_i,a_{-i}^{(\tau,s)})}{B}}|0\rangle \tag{22}$$

$$+ \sqrt{1-\frac{\mathcal{L}_i(a_i,a_{-i}^{(\tau,s)})}{B}}|1\rangle\right)$$

$$= |k\rangle|i\rangle|a_i\rangle \frac{1}{\sqrt{h_{k,t}H^{k-1}}} \sqrt{\sum_{\tau=(r_{k,t}-1)H^k+1}^{(r_{k,t}-1)H^k+h_{k,t}H^{k-1}} \frac{\ell_{i,\tau}(a_i)}{B}}|\psi_i\rangle|0\rangle + |\phi_i\rangle|1\rangle, \tag{23}$$

where $|\psi_i\rangle$ is a normalized state and $|\phi_i\rangle$ is an unnormalized garbage state, and for any $i \in [m]$, $\langle k|\langle i|V_t|k\rangle|i\rangle$ is a $Bh_{k,t}H^{k-1}$-amplitude-encoding of the vector $\bar{\ell}_{k,t} := \left(\sum_{\tau=(r_{k,t}-1)H^k+1}^{(r_{k,t}-1)H^k+h_{k,t}H^{k-1}} \ell_{i,\tau}(a_i)\right)_{a_i \in \mathcal{A}_i}$. Given the amplitude-encoding of $\bar{\ell}_{k,t}$, we can implement the Gibbs sampling oracle $\mathcal{O}^{\text{Gibbs}}_{\sqrt{\log n/H}\bar{\ell}_{k,t}}(\delta)$ using Theorem 6.

After $T$ rounds, we prepare the uniform superposition of $t \in [T]$ and $k \in [2^K]$. By coherently apply $V_t$ controlled by an ancilla register $|t\rangle$, we can implement $\sum_{t\in[T]}|t\rangle\langle t| \otimes V_t$. Then following the previous steps, we can apply $\mathcal{O}^{\text{Gibbs}}_{\sqrt{\log n/H}\bar{\ell}_{k,t}}(\delta)$ conditioning on the first two registers containing $|t\rangle|k\rangle$. The resulting state is the output of Algorithm 1.

## B.2  Proof of Theorem 1

We give the formal version of Theorem 1 and provide its proof.

**Theorem 8.** *For any $m$-player normal-form game with $n$ actions for each player and $\alpha \in (0,1)$, Algorithm 1 outputs an $\varepsilon$-correlated equilibrium of the game with success probability at least $1 - \alpha$ using $O(m\sqrt{n}\log(1/\alpha)) \cdot \left(\log(n)B/\varepsilon\right)^{O(B/\varepsilon)} \cdot \text{poly}(\log n, \log m, 1/\varepsilon, B)$ queries to $\mathcal{O}_\mathcal{L}$ and $O(m^2\sqrt{n}\log(1/\alpha)) \cdot \left(\log(n)B/\varepsilon\right)^{O(B/\varepsilon)} \cdot \text{poly}(\log n, \log m, 1/\varepsilon, B)$ time.*

*Proof.* **Correctness.** At the $t$-th round, denote the output distribution of the quantum Gibbs sampler $\mathcal{O}^{\text{Gibbs}}_{\sqrt{\log n/H}\bar{\ell}_{k,t}}(\delta)$ by $\tilde{q}_{i,k,t}$ and $\tilde{p}_{i,t} := \frac{1}{2^K}\sum_{k\in[2^K]}\tilde{q}_{i,k,t}$. By Theorem 6, we have $\|\tilde{q}_{i,k,t}-q_{i,k,t}\|_1 \le \delta$ and hence $\|\tilde{p}_{i,t}-p_{i,t}\|_1 \le \delta$. The output state of Algorithm 1 is

$$|\psi_o\rangle = \frac{1}{\sqrt{T2^K}} \sum_{t=1}^{T}\sum_{k=1}^{2^K} |t\rangle|k\rangle \bigotimes_{i\in[m]} \left(\sum_{a_i\in\mathcal{A}_i} \sqrt{\tilde{q}_{i,k,t}(a_i)}|a_i\rangle_{A_i}|\psi_{a_i}\rangle\right), \tag{24}$$

which can be written as

$$\frac{1}{\sqrt{T}} \sum_{t \in [T]} \bigotimes_{i \in [m]} \Big( \sum_{a_i \in \mathcal{A}_i} \sqrt{\tilde{p}_{i,t}(a_i)} |a_i\rangle_{A_i} |\phi_{a_i}\rangle \Big) \tag{25}$$

for some normalized states $|\phi_{a_i}\rangle$. Measuring the register $A_i$ for all $i \in [m]$ gives the distribution $\frac{1}{T} \sum_{t \in [T]} \otimes_{i \in [m]} \tilde{p}_{i,t}$.

For any player $i \in [m]$, since $p_{i,t}$ is the strategy of the multi-scale MWU algorithm with parameters $K = \log_2(3B/\varepsilon) + 1$, $H = 4 \log(n) 2^{2K}$, $T = H^{2^K}$ at the $t$-th round given loss vectors $\ell_{i,1}, \ldots, \ell_{i,t-1}$, by Theorem 5, we have

$$\frac{1}{T} \sum_{t=1}^{T} \langle p_{i,t}, \ell_{i,t} \rangle - \frac{1}{T} \min_{\phi \in \Phi_i} \sum_{t=1}^{T} \sum_{j=1}^{n} p_{i,t}(j) \cdot \ell_{i,t}(\phi(j)) \leq \frac{\varepsilon}{3B} B = \frac{\varepsilon}{3}. \tag{26}$$

Let $\tilde{\ell}_{i,t} := \mathcal{L}(\cdot, \tilde{p}_{-i,t})$ be the expected loss vector of player $i$ in the $t$-th round. Since $\ell_{i,t}$ is the average of $\mathcal{L}(\cdot, a_{-i}^{(t,s)})$ for $s \in [S]$ and $a_{-i}^{(t,s)}$ is sampled independently from $\tilde{p}_{-i,t}$, by Hoeffding's inequality, we have

$$\Pr\left[ |\ell_{i,t}(a_i) - \tilde{\ell}_{i,t}(a_i)| \geq \frac{\varepsilon}{6} \right] \leq 2 \exp\left(-\frac{\varepsilon^2 S}{18 B^2}\right) \leq \frac{\alpha}{mnT}. \tag{27}$$

for any $i \in [m]$, $t \in [T]$, and $a_i \in \mathcal{A}_i$. Taking a union bound over $i \in [m]$, $t \in [T]$, and $a_i \in \mathcal{A}_i$, we have

$$|\ell_{i,t}(a_i) - \tilde{\ell}_{i,t}(a_i)| \leq \frac{\varepsilon}{6} \tag{28}$$

for all $i \in [m]$, $t \in [T]$, and $a_i \in \mathcal{A}_i$ with probability at least $1 - \alpha$. Therefore, for any swap function $\phi \in \Phi_i$, we have

$$\frac{1}{T} \sum_{t=1}^{T} \langle p_{i,t}, \tilde{\ell}_{i,t} \rangle - \frac{1}{T} \sum_{t=1}^{T} \sum_{j=1}^{n} p_{i,t}(j) \cdot \tilde{\ell}_{i,t}(\phi(j)) \tag{29}$$

$$\leq \frac{1}{T} \sum_{t=1}^{T} \langle p_{i,t}, \ell_{i,t} \rangle - \frac{1}{T} \sum_{t=1}^{T} \sum_{j=1}^{n} p_{i,t}(j) \cdot \ell_{i,t}(\phi(j)) + \frac{\varepsilon}{3} \tag{30}$$

$$\leq \frac{\varepsilon}{3} + \frac{\varepsilon}{3} = \frac{2\varepsilon}{3}. \tag{31}$$

Let $\mathcal{D}$ be the distribution $\frac{1}{T} \sum_{t \in [T]} \otimes_{i \in [m]} \tilde{p}_{i,t}$. Since $p_{i,t}, \tilde{p}_{i,t}$ is the uniform mixture of $q_{i,k,t}, \tilde{q}_{i,k,t}$ for $k \in [2^K]$ respectively and $\|\tilde{q}_{i,k,t} - q_{i,k,t}\|_1 \leq \delta$, we have $\|\tilde{p}_{i,t} - p_{i,t}\|_1 \leq \delta$. Then we have

$$\mathbb{E}_{a \sim \mathcal{D}}[\mathcal{L}_i(a_i, a_{-i})] - \mathbb{E}_{a \sim \mathcal{D}}[\mathcal{L}_i(\phi(a_i), a_{-i})] \tag{32}$$

$$= \frac{1}{T} \sum_{t=1}^{T} \langle \tilde{p}_{i,t}, \tilde{\ell}_{i,t} \rangle - \frac{1}{T} \sum_{t=1}^{T} \sum_{j=1}^{n} \tilde{p}_{i,t}(j) \cdot \tilde{\ell}_{i,t}(\phi(j)) \tag{33}$$

$$\leq \frac{1}{T} \sum_{t=1}^{T} \langle p_{i,t}, \tilde{\ell}_{i,t} \rangle - \frac{1}{T} \sum_{t=1}^{T} \sum_{j=1}^{n} p_{i,t}(j) \cdot \tilde{\ell}_{i,t}(\phi(j)) + 2\delta B \tag{34}$$

$$\leq \frac{2\varepsilon}{3} + \frac{\varepsilon}{3} = \varepsilon. \tag{35}$$

Therefore, $\mathcal{D}$ is an $\varepsilon$-CE of the game.

**Query complexity.** Each call to $V_t$ in Eq. (22) requires one query $\mathcal{O}_{\mathcal{L}}$ and $\mathcal{O}_{\mathcal{L}}^{\dagger}$. By Theorem 6, we need $\tilde{O}(Bh_{k,t}H^{k-1} \cdot \sqrt{\log n/H} \cdot \sqrt{n})$ calls to $V_t$ to get a sample from $\tilde{q}_{i,k,t}$. Since $h_{k,t}$ is smaller than $H$ and $H^k \leq H^{2^K} = T$ for $k \in [2^K]$, we need

$$\tilde{O}(Bh_{k,t}H^{k-1} \cdot \sqrt{\log n/H} \cdot \sqrt{n}) = \tilde{O}(BH^{k-1/2}\sqrt{n}) = \tilde{O}(BT\sqrt{n}) \tag{36}$$

calls to $V_t$ to sample from $\tilde{q}_{i,k,t}$. Since we need to get $S$ samples for $m$ players in $T$ rounds, the total query complexity is

$$\tilde{O}(BT\sqrt{n} \cdot S \cdot m \cdot T) = \tilde{O}(T^2 m\sqrt{n}\log(1/\alpha)), \tag{37}$$

where the $\tilde{O}$ notation hides polynomial factors in $\log n, \log m, 1/\varepsilon, B$. Substituting $T = H^{2^K} = \left(\log(n)B/\varepsilon\right)^{O(B/\varepsilon)}$, the query complexity is

$$O(m\sqrt{n}\log(1/\alpha)) \cdot \left(\log(n)B/\varepsilon\right)^{O(B/\varepsilon)} \cdot \mathrm{poly}(\log n, \log m, 1/\varepsilon, B). \tag{38}$$

**Time complexity.** There are $TS$ entries in the QRAM and each entry has $m\log n$ bits, so the time complexity of applying $U_{\mathrm{QRAM}}$ and modifying one entry is $O(TSm\log n)$ [5]. At each round, we need to modify $S$ entries of the QRAM to store the new samples. To prepare and sample from the Gibbs state, we need to call $U_{\mathrm{QRAM}}$ the same times as the number of queries to $\mathcal{O}_\mathcal{L}$. Therefore, the time complexity is

$$\left(TS + O(m\sqrt{n}\log(1/\alpha)) \cdot \left(\log(n)B/\varepsilon\right)^{O(B/\varepsilon)} \cdot \mathrm{poly}(\log n, \log m, 1/\varepsilon, B)\right) \cdot O(TSm\log n)$$

$$=O(m^2\sqrt{n}\log^2(1/\alpha)) \cdot \left(\log(n)B/\varepsilon\right)^{O(B/\varepsilon)} \cdot \mathrm{poly}(\log n, \log m, 1/\varepsilon, B). \tag{39}$$

$\square$

### B.3 Implementation of Algorithm 2

In this section we introduce our Gibbs sampling method in Algorithm 2. Specifically, we extend the dynamic Gibbs sampling of two-player games, as given in Lemma 2, to multi-player games, and provide a more refined explanation of the query and gate complexity.

**Lemma 2** (Theorem 3 in Bouland et al. [9]). *For failure probability $\alpha \in (0,1)$ and $\delta < \eta$, given a quantum oracle for $A \in \mathbb{R}^{n_1 \times n_2}$ with $\|A\|_{\max} \leq 1$, there is a quantum algorithm which solves Problem 2 with probability more than $1 - \alpha$ using*

$$\max(\mathcal{T}_{\mathrm{samp}}, \mathcal{T}_{\mathrm{update}}) = O\left(1 + \sqrt{n_1}T\eta \cdot \log^4\left(\frac{n_1 n_2}{\delta}\right)\left(\sqrt{\eta\log\left(\frac{n_1\eta T}{\alpha}\right)} + \eta\log\left(\frac{n_1\eta T}{\alpha}\right)\right)\right)$$

*time with an additive initialization cost of $O\left(\eta^3 T^3 \log^4\left(\frac{n_1\eta T}{\delta}\right) + \log^7\left(\frac{n_1\eta T}{\delta}\right)\right)$.*

The complexity of this method consists of two parts: maintaining the data structure in each round and sampling from it. It should be noted that here we assume access to a classical-write / quantum-read random access memory at unit cost. In the actual implementation, if we consider the gate complexity of QRAM, we need additional gate complexity, which is proportional to the number of entries in QRAM and the number of bits per entry.

The Gibbs sampling used in Algorithm 2 can be formalized as Problem 3, which is an $m$-player game version of Problem 2. Note that the vector $x$ of size $n$ maintained in Problem 2 actually records and maintains the combination of opponent's strategies of the previous $t$ rounds, where in each round one particular action is updated. In $m$-player games, the $m-1$ opponents have $n^{m-1}$ possible strategies, and we can use a high-dimensional array to maintain the information of opponent strategies. A simple idea is that we can use the method in Bouland et al. [9] to store the opponents' strategies in the high-dimensional array of size $n^{m-1}$ using a special data structure called "sampler tree", but the cost would be exponential large in storage space, leading to exponential gate complexity if using QRAM. Considering that the array is sparse, we improved this method by using QRAM to directly store the strategies from each round, achieving better time complexity.

**Problem 3** (Sampling maintenance for $m$-player game). *Given $\eta > 0$, $0 < \delta < 1$, and suppose that we have a quantum oracle for the loss function $\mathcal{L}_i(j, a_{-i}^{(t)}) \in [0, B]$. For player $i$, consider a sequence of size $T$, where each item includes an "Update" operation to $(m-1)$-dimension dynamic arrays $D$ indexed by actions of the $m-1$ opponents' strategies $x_{-i} = (x_1, x_2, \ldots, x_{i-1}, x_{i+1}, \ldots, x_{m-1})$ where $x_j \in [n]$, with each entry $D_{(x_{-i})} \geq 0$. Each "Update" operation takes the form of $D_{(x_{-i})} \leftarrow D_{(x_{-i})} + \eta$ for some $D_{(x_{-i})} \in [n]^{m-1}$. Let $\mathcal{T}_{\mathrm{update}}$ denote queries per operation we need to maintain a $\delta$-approximate Gibbs oracle $\mathcal{O}_{\mathcal{L}_i(j,D)}^{\mathrm{dynamic\text{-}Gibbs}}$ of vector $\mathcal{L}_i(j, D)$ (for different strategies $j$), and let $\mathcal{T}_{\mathrm{samp}}$ denote time needed for $\mathcal{O}_{\mathcal{L}_i(j,D)}^{\mathrm{dynamic\text{-}Gibbs}}$.*

In the algorithm proposed by Bouland et al. [9], a key step involves using sampler tree to store $x \in \mathbb{R}^n_{\geq 0}$ and prepare a $t\eta$-amplitude encoding of $Ax$ (Corollary 4 in [9]):

$$\mathcal{O}_{Ax}|0\rangle|0\rangle|j\rangle = |0\rangle \left( \sum_{i\in[n]} \sqrt{\frac{A_{ij}x_i}{\beta}}|0\rangle|i\rangle + |1\rangle|g\rangle \right) |j\rangle, \text{ here } \beta \geq \|x\|_1 \text{ and } |g\rangle \text{ is unnormalized.}$$
(40)

Corollary 4 in [9] shows that we can maintain the oracle $\mathcal{O}_{Ax}$ with total building time cost $O(T \log n)$ after $T$ rounds, and each call of $\mathcal{O}_{Ax}$ requires $O(\log n)$ time and $O(1)$ queries to the given oracle $\mathcal{O}_{\mathcal{L}}$. However, this is based on the assumption of access to a classical-write / quantum-read random access memory at unit cost. For gate complexity, such an assumption neglects the entries of this data structure ($n$ for two-player games) and the number of bits used to store information, which is related to the precision we require.

Instead of maintaining the sampler tree in Bouland et al. [9], we maintain a QRAM storing the sample of strategies, which means that at time $t$, we can access the unitary $U_{\text{QRAM}}$ such that

$$U_{\text{QRAM}}|\tau\rangle|0\rangle \mapsto |\tau\rangle|a^{(\tau)}\rangle$$
(41)

for all $\tau < t$, where $a^\tau \in \mathcal{A}$ is the sampler at time $\tau$. Accordingly, in our algorithm we need to implement a $t$-amplitude encoding of

$$\sum_{\tau=1}^{t} \mathcal{L}_i(\cdot, a^{(\tau)}_{-i}).$$
(42)

This is can be implemented by performing

$$|a_i\rangle|0\rangle \mapsto |a_i\rangle \frac{1}{\sqrt{t}} \sum_{\tau=1}^{t} |\tau\rangle|a^{(\tau)}_i\rangle|a^{(\tau)}_{-i}\rangle \left( \sqrt{\frac{\mathcal{L}_i(a_i, a^{(\tau)}_{-i})}{B}}|1\rangle + \sqrt{1 - \frac{\mathcal{L}_i(a_i, a^{(\tau)}_{-i})}{B}}|0\rangle \right)$$
(43)

$$= |a_i\rangle \left( \sqrt{\frac{1}{tB} \sum_{\tau=1}^{t} \mathcal{L}_i(a_i, a^{(\tau)}_{-i})} |\psi_i\rangle|1\rangle + |\phi_i\rangle|0\rangle \right)$$
(44)

for some normalized state $|\psi_i\rangle$ and unnormalized state $|\phi_i\rangle$. This is a $tB$-amplitude encoding of $\sum_{\tau=1}^{t} \mathcal{L}_i(\cdot, a^{(\tau)}_{-i})$.

There are $T$ entries in the QRAM. For the precision $\delta$ to be considered, each entry has $O(m \log n)$ bits, and thus the gate complexity of applying one $U_{\text{QRAM}}$ and modifying one entry is $O(Tm \log n)$. Note that if we also use a sampler tree to directly store the sparse high-dimensional array $D$, since $D$ has $n_2 = n^{m-1}$ entries, we will similarly require $\tilde{O}(\log n_2) = \tilde{O}(m \log n)$ queries to the sampler tree. However, the additional cost is that the sampler tree itself requires exponentially large storage space, and thus leads to an exponential gate complexity if using QRAM for storage. For query complexity, both construction methods require $O(1)$ queries to achieve $t$-amplitude encoding of $\sum_{\tau=1}^{t} \mathcal{L}_i(\cdot, a^{(\tau)}_{-i})$.

Here we present a modified version of Theorem 3 in Bouland et al. [9]:

**Lemma 3** (modified version of Lemma 2 for $m$-player game). *Let $n_2 := n^{m-1}$. For failure probability $\alpha \in (0,1)$ and $\delta < \eta$, given a quantum oracle $\mathcal{O}_{\mathcal{L}}$, there is a quantum algorithm which solves Problem 3 with probability more than $1 - \alpha$ using*

$$\max(\mathcal{T}_{\text{samp}}, \mathcal{T}_{\text{update}})$$

$$= O\left( 1 + \sqrt{n} \cdot T\eta \cdot Tm \log n \cdot \log^4\left(\frac{n}{\delta}\right) \cdot \left( \sqrt{\eta \log\left(\frac{n\eta T}{\alpha}\right)} + \eta \log\left(\frac{n\eta T}{\alpha}\right) \right) \right)$$

*quantum gates and*

$$O\left( 1 + \sqrt{n} \cdot T\eta \cdot \log^4\left(\frac{n}{\delta}\right) \cdot \left( \sqrt{\eta \log\left(\frac{n\eta T}{\alpha}\right)} + \eta \log\left(\frac{n\eta T}{\alpha}\right) \right) \right)$$

*queries to $\mathcal{O}_{\mathcal{L}}$, with an additive initialization cost of $O\left( \eta^3 T^3 \log^4\left(\frac{n\eta T}{\delta}\right) + \log^7\left(\frac{n\eta T}{\delta}\right) \right)$.*

The proof of the lemma is entirely consistent with Lemma 2, where we simply use the aforementioned QRAM to replace the sampler tree to implement the $t$-amplitude encoding of $\sum_{\tau=1}^{t} \mathcal{L}_i(\cdot, a_{-i}^{(\tau)})$. We only need to make slight modifications to the parameters, as noted in Remark 1.

**Remark 1.** *In the results presented in [9], the term related to the number of opponent strategies $n_2 = n^{m-1}$ is of the form $\log^4 n_2$. However, in their sampling algorithm, the authors only used $O(\log n_2)$ queries to the sampler tree to prepare an oracle $\mathcal{O}_{Ax}$ within the sampler tree. There are no computations involving time that are dependent on $n_2$ in the other steps. Hence, this term can actually be corrected to $\log^1 n_2$, which corresponds to the time of achieve $t$-amplitude encoding of $\sum_{\tau=1}^{t} \mathcal{L}_i(\cdot, a_{-i}^{(\tau)})$. By replacing the sampler tree with QRAM, we obtain our gate complexity with the term $Tm \log n$, as showed in Lemma 3. Furthermore, as we only need $O(1)$ queries of $\mathcal{O}_{\mathcal{L}}$ to achieve the encoding, the query complexity does not include the term $Tm \log n$.*

**Remark 2.** *The complexity in [9] has an additive $\epsilon^{-3}$ term, which arises from an additive initialization cost $\tilde{O}(\eta^3 T^3)$ in Lemma 2. This term is unrelated to the number of queries to the loss oracle $\mathcal{O}_{\mathcal{L}}$ and appears only in the time complexity. The distinction is that their QRAM model assumes that mathematical operations can be implemented exactly in $O(1)$ time, whereas we further consider the gate complexity of QRAM operations in our analysis. When considering query complexity, their dependence on $\epsilon$ is $\tilde{O}(1/\varepsilon^{2.5})$, which matches ours exactly. However, for the time complexity, due to our additional consideration of the gate complexity of QRAM operations, our overall time complexity becomes $\tilde{O}(1/\varepsilon^{4.5})$, which is larger than $\varepsilon^{-3}$. Therefore, we do not explicitly include the additive initialization cost term $\varepsilon^{-3}$ in the final stated result.*

## B.4 Proof of Theorem 2

In this subsection, we will provide a proof showing that Algorithm 2 can output an $\varepsilon$-coarse correlated equilibrium with high probability, and calculate the complexity based on the results in Lemma 3. The formal version of Theorem 2 is stated below:

**Theorem 9.** *For any $m$-player normal-form game with $n$ actions for each player and $\alpha \in (0, 1)$, Algorithm 2 computes an $\varepsilon$-coarse correlated equilibrium of the game with success probability at least $1 - \alpha$ using $\tilde{O}(mn^{\frac{1}{2}} B^{\frac{5}{2}} \varepsilon^{-\frac{5}{2}})$ queries to $\mathcal{O}_{\mathcal{L}}$ and $\tilde{O}(m^2 n^{\frac{1}{2}} B^{\frac{9}{2}} \varepsilon^{-\frac{9}{2}})$ time.*

*Proof.* **Correctness.** For convenience, we denote $s_i^{(t)}$ by the vector for Gibbs sampling of player $i$ in $t$-th round, i.e., $s_i^{(t)} := -\eta \cdot \sum_{k=0}^{t-1} \mathcal{L}(j, a_{-i}^{(k)})$. The proof of the correctness of Theorem 9 consists of two main parts: First, we demonstrate that the uniform mixture of Gibbs distribution of $s_i^{(t)}$ in each round is an $O(\varepsilon)$-coarse correlated equilibrium of this normal-form game. Then we consider the action strategies $a_i^{(t)}$ generated by the Gibbs sampling in our algorithm, and we will show that they can also derive an approximate coarse correlated equilibrium.

Denote $u_i^{(t)} := \frac{\exp(s_i^{(t)})}{\|\exp(s_i^{(t)})\|_1}$ and $\ell_i^{(t)} := \mathcal{L}_i(\cdot, a_{-i}^{(t)})$ for all $t = 0, \ldots, T-1$. The regret bound of MWU (Theorem 4) implies that

$$\sum_{t=0}^{T-1} \langle u_i^{(t)}, \ell_i^{(t)} \rangle - \sum_{t=0}^{T-1} \langle u, \ell_i^{(t)} \rangle \leq 2B\sqrt{\log(n)T} \tag{45}$$

for all $i \in [m]$ and $u \in \Delta([n])$.

We now use a "ghost iteration" argument in [9] to bound the regret of $u_i^{(t)}$ with respect to loss vectors $\hat{\ell}_i^{(t)} := \mathcal{L}_i(\cdot, u_{-i}^{(t)})$. Denote $\tilde{\ell}_i^{(t)} := \hat{\ell}_i^{(t)} - \ell_i^{(t)}$, $\tilde{u}_i^{(0)} := u_i^{(0)}$, and

$$\tilde{u}_i^{(t)} = \frac{\exp(-\eta \sum_{\tau=0}^{t-1} \tilde{\ell}_i^{(\tau)})}{\|\exp(-\eta \sum_{\tau=0}^{t-1} \tilde{\ell}_i^{(\tau)})\|_1} \tag{46}$$

for $t = 1, \ldots, T-1$. Then Theorem 4 again implies that

$$\sum_{t=0}^{T-1} \langle \tilde{u}_i^{(t)}, \tilde{\ell}_i^{(t)} \rangle - \sum_{t=0}^{T-1} \langle u, \tilde{\ell}_i^{(t)} \rangle \leq 2B\sqrt{\log(n)T} \tag{47}$$

for all $i \in [m]$ and $u \in \Delta([n])$.

Summing Eq. (45) and Eq. (47) gives us

$$\sum_{t=0}^{T-1} \langle u_i^{(t)}, \hat{\ell}_i^{(t)} \rangle - \sum_{t=0}^{T-1} \langle u, \hat{\ell}_i^{(t)} \rangle + \sum_{t=0}^{T-1} \langle \tilde{u}_i^{(t)} - u_i^{(t)}, \hat{\ell}_i^{(t)} - \ell_i^{(t)} \rangle \leq 4B\sqrt{\log(n)T}. \qquad (48)$$

Considering that $u$ can be arbitrarily chosen in $\Delta([n])$, we have

$$\max_{u \in \Delta([n])} \left\{ \sum_{t=0}^{T-1} \langle u_i^{(t)}, \hat{\ell}_i^{(t)} \rangle - \sum_{t=0}^{T-1} \langle u, \hat{\ell}_i^{(t)} \rangle \right\} + \sum_{t=0}^{T-1} \langle \tilde{u}_i^{(t)} - u_i^{(t)}, \hat{\ell}_i^{(t)} - \ell_i^{(t)} \rangle \leq 4B\sqrt{\log(n)T}. \quad (49)$$

Taking the expectation of the left-hand side, we have

$$\mathbb{E}\left[ \max_{u \in \Delta([n])} \left\{ \sum_{t=0}^{T-1} \langle u_i^{(t)}, \hat{\ell}_i^{(t)} \rangle - \sum_{t=0}^{T-1} \langle u, \hat{\ell}_i^{(t)} \rangle \right\} \right] + \mathbb{E}\left[ \sum_{t=0}^{T-1} \langle \tilde{u}_i^{(t)} - u_i^{(t)}, \hat{\ell}_i^{(t)} - \ell_i^{(t)} \rangle \right]$$
$$\leq 4B\sqrt{\log(n)T}. \qquad (50)$$

Consider the second term on the left-hand side,

$$\mathbb{E}_{a^{(0)},\cdots,a^{(t)}}\left[ \langle \tilde{u}_i^{(t)} - u_i^{(t)}, \hat{\ell}_i^{(t)} - \ell_i^{(t)} \rangle \right] = \mathbb{E}_{a^{(0)},\cdots,a^{(t-1)}}\left[ \langle \tilde{u}_i^{(t)} - u_i^{(t)}, \hat{\ell}_i^{(t)} - \mathbb{E}_{a^{(t)}}[\ell_i^{(t)}] \rangle \right]. \quad (51)$$

Suppose that the Gibbs sampling oracle gives $a_i^{(t)}$ from $\tilde{p}_i^{(t)}$, by the assumption $\|\tilde{p}_i^{[t]} - u_i^{(t)}\|_1 \leq \delta$, we have $\| \bigotimes_{j \neq i} \tilde{p}_j^{(t)} - \bigotimes_{j \neq i} u_j^{(t)} \|_1 \leq (n-1)\delta$. Note that $\mathbb{E}_{a^{(t)}}[\ell_i^{(t)}] = \mathcal{L}_i(\cdot, \tilde{p}_{-i}^{(t)})$, as $\mathcal{L}_i \in [0, B]$, we have

$$\mathbb{E}\left[ \sum_{t=0}^{T-1} \langle u_i^{(t)} - \tilde{u}_i^{(t)}, \hat{\ell}_i^{(t)} - \ell_i^{(t)} \rangle \right] = \sum_{t=0}^{T-1} \mathbb{E}\left[ \langle u_i^{(t)} - \tilde{u}_i^{(t)}, \hat{\ell}_i^{(t)} - \ell_i^{(t)} \rangle \right] \leq 2(n-1)BT\delta. \qquad (52)$$

Therefore, summing Eq. (50) and Eq. (52), taking $T \geq \frac{64B^2 \log n}{\varepsilon^2}$ and $\delta \leq \frac{\varepsilon}{8(n-1)B}$, for $\bar{u} = (\bar{u}_1, \ldots \bar{u}_m) := \frac{1}{T} \sum_{t=0}^{T-1} (u_1^{(t)}, u_1^{(t)}, \cdots u_m^{(t)})$, we have

$$\mathbb{E}_{u_1, u_2, \ldots u_T} \left[ \max_{a_i' \in \Delta([n])} \left\{ \mathbb{E}_{a \sim \bar{u}}[\mathcal{L}_i(a_i, a_{-i})] - \mathbb{E}_{a \sim \bar{u}}[\mathcal{L}_i(a_i', a_{-i})] \right\} \right]$$

$$\leq \mathbb{E}_{u_1, u_2, \ldots u_T} \left[ \frac{1}{T} \max_{u \in \Delta([n])} \left\{ \sum_{t=0}^{T-1} \langle u_i^{(t)}, \hat{\ell}_i^{(t)} \rangle - \sum_{t=0}^{T-1} \langle u, \hat{\ell}_i^{(t)} \rangle \right\} \right] \qquad (53)$$

$$\leq \frac{\varepsilon}{2}. \qquad (54)$$

Next, by a martingale argument we will prove that with high probability, Algorithm 2 implicitly provides an $\varepsilon$-coarse correlated equilibrium $\bar{u}$.

Consider a filtration given by $\mathcal{F}_t = \sigma(s^{(0)}, s^{(1)}, \cdots s^{(t)})$, where $s^{(t)} := (s_1^{(t)}, s_2^{(t)}, \cdots s_m^{(t)})$. Define a martingale sequence of the form $D_t := \langle u_i^{(t)} - \tilde{u}_i^{(t)}, \hat{\ell}_i^{(t)} - \ell_i^{(t)} \rangle - \langle \tilde{u}_i^{(t)} - u_i^{(t)}, \hat{\ell}_i^{(t)} - \mathbb{E}[\ell_i^{(t)} | \mathcal{F}_{t-1}] \rangle$. Notice that with probability 1 we have $|D_t| \leq 4B$. Azuma's inequality implies that

$$\Pr[\sum_{t=0}^{T-1} D_t \geq \frac{\varepsilon}{4}T] \leq \exp\left( \frac{-(\varepsilon T/4)^2}{2T \cdot (4B)^2} \right) = \exp\left( \frac{-\varepsilon^2 T}{512B^2} \right). \qquad (55)$$

Taking $T \geq \frac{512B^2 \log \frac{4}{\alpha}}{\varepsilon^2}$, we thus have

$$\sum_{t=0}^{T-1} \langle u_i^{(t)} - \tilde{u}_i^{(t)}, \hat{\ell}_i^{(t)} - \ell_i^{(t)} \rangle \leq \sum_{t=0}^{T-1} \mathbb{E}\left[ \langle u_i^{(t)} - \tilde{u}_i^{(t)}, \hat{\ell}_i^{(t)} - \ell_i^{(t)} \rangle \right] + \frac{\varepsilon}{4}T \qquad (56)$$

with probability more than $1 - \frac{\alpha}{4}$.

Combining Eq. (49) with Eq. (52), it gives us

$$\max_{a_i' \in \Delta([n])} \left\{ \mathbb{E}_{a \sim \bar{u}}[\mathcal{L}_i(a_i, a_{-i})] - \mathbb{E}_{a \sim \bar{u}}[\mathcal{L}_i(a_i', a_{-i})] \right\} = \max_{u \in \Delta([n])} \left\{ \sum_{t=0}^{T-1} \langle u_i^{(t)}, \ell_i^{(t)} \rangle - \sum_{t=0}^{T-1} \langle u, \ell_i^{(t)} \rangle \right\}$$

$$\leq \frac{3\varepsilon}{4} \tag{57}$$

with probability at least $1 - \frac{\alpha}{4}$. That is to say, $\bar{u}$ is an $O(\varepsilon)$-coarse correlated equilibrium with probability at least $1 - \frac{\alpha}{4}$.

Finally, note that Gibbs sampling implicitly implements the sampling oracles for $u_i^{(t)}$, but cannot directly provide these distribution vectors explicitly. We will prove that a coarse correlated equilibrium (i.e., $\hat{x}_i$) can be found with probability at least $1 - \alpha$ based on $a_i^{(t)}$ from Gibbs sampling in each round.

We previously used the notation $\tilde{p}_i^{(t)}$ to represent the actual distribution of $a_i^{(t)}$ sampled from Gibbs sampling. Let $\bar{p}_i := \frac{1}{T} \sum_{t=0}^{T-1} \tilde{p}_i^{(t)}$. Since $\|\tilde{p}_i^{[t]} - u_i^{(t)}\|_1 \leq \delta$, by the convexity of norms we have $\|\bar{p}_i - \bar{u}_i\|_1 \leq \delta$, thus for any action $a_i'$ of player $i$, loss of player $i$ under the two different opponent strategies is nearly the same:

$$\left| \mathbb{E}_{a \sim \tilde{p}}[\mathcal{L}_i(a_i', a_{-i})] - \mathbb{E}_{a \sim \bar{u}}[\mathcal{L}_i(a_i', a_{-i})] \right| \leq (n-1)B\delta. \tag{58}$$

For a fixed strategy $a_i'$ of player $i$, let random variable $X_j$ denote player $i$'s loss when sampling the opponent's strategy $a_{-i}$ from distribution $\tilde{p}_{-i}^{(j)}$. Thus $X_t \in [0, B]$ and $\mathbb{E}(X_t) = \mathbb{E}_{a \sim \tilde{p}^{(t)}}[\mathcal{L}_i(a_i', a_{-i})] \in [0, B]$. Note that $S_t := \sum_{j=0}^{t-1}(X_j - E[X_j])$ is a martingale sequence generated by filtration $\mathcal{F}$. Again by Azuma's inequality,

$$\Pr\left[ |S_T - S_0| \geq \frac{\varepsilon}{16}T \right] \leq 2\exp\left( -\frac{(\varepsilon T/16)^2}{2 \cdot T \cdot B^2} \right) = 2\exp\left( -\frac{T\varepsilon^2}{512B^2} \right). \tag{59}$$

This implies that

$$\Pr\left[ |\mathcal{L}_i(a_i', \hat{x}_{-i}) - \mathbb{E}_{a \sim \tilde{p}}[\mathcal{L}_i(a_i', a_{-i})]| \leq \frac{\varepsilon}{16} \right] = \Pr\left[ \frac{1}{T} \left| \sum_{t=0}^{T-1} X_t - \sum_{t=0}^{T-1} \mathbb{E}[X_t] \right| \geq \frac{\varepsilon}{16} \right]$$

$$\leq 2\exp\left( -\frac{T\varepsilon^2}{512B^2} \right). \tag{60}$$

Take $\delta \leq \frac{\varepsilon}{16B(n-1)}$ and $T \geq \frac{512B^2 \log(4/\alpha)}{\varepsilon^2}$, combining Eq. (58) and Eq. (60), with probability at least $1 - \frac{\alpha}{2}$ we have

$$\left| \mathcal{L}_i(a_i', \hat{x}_{-i}) - \mathbb{E}_{a \sim \bar{u}}[\mathcal{L}_i(a_i', a_{-i})] \right| \leq \frac{\varepsilon}{8}, \tag{61}$$

for any strategy $a_i'$.

Summing Eq. (57) and Eq. (61), we have with success probability at least $(1 - \frac{\alpha}{4}) \cdot (1 - \frac{\alpha}{4}) \geq 1 - \alpha$,

$$\left| \mathbb{E}_{a \sim \hat{x}}[\mathcal{L}_i(a_i, a_{-i})] - \mathbb{E}_{a \sim \hat{x}}[\mathcal{L}_i(a_i', a_{-i})] \right| \leq \varepsilon,$$

which means the output of Algorithm 2 forms an $\varepsilon$-coarse correlated equilibrium for the normal-form game.

**Query complexity.** In each round, $m$ Gibbs samplings are required, corresponding to $m$ instances of Problem 3. According to the Lemma 3, each sampling requires $\tilde{O}(\sqrt{n} \cdot T\eta^{\frac{3}{2}}) = \tilde{O}(\sqrt{n}B^{\frac{1}{2}}\varepsilon^{-\frac{1}{2}})$. Therefore, the total query complexity is $\tilde{O}(T \cdot \sqrt{n}mB\varepsilon^{-\frac{1}{2}}) = \tilde{O}(n^{\frac{1}{2}}mB^{\frac{5}{2}}\varepsilon^{-\frac{5}{2}})$.

**Time complexity.** By Lemma 3, each sampling takes time $\tilde{O}(\sqrt{n} \cdot T\eta^{\frac{3}{2}} \cdot Tm\log n) = \tilde{O}(\sqrt{n}mB^{\frac{5}{2}}\varepsilon^{-\frac{5}{2}})$. The total time complexity is $\tilde{O}(T \cdot m \cdot \sqrt{n}mB^{\frac{1}{2}}\varepsilon^{-\frac{1}{2}}) + \tilde{O}(\eta^3 T^3) = \tilde{O}(n^{\frac{1}{2}}m^2B^{\frac{9}{2}}\varepsilon^{-\frac{9}{2}})$. $\qquad\square$

**Remark 3.** *Note that by replacing the quantum Gibbs sampling with exact Gibbs sampling oracles, we can follow the correctness proof above and derive a classical query complexity of $\tilde{O}(mn/\varepsilon^2)$, as is shown in Corollary 1.*

## C   Technical details of lower bounds

In this appendix, we present the formal proofs of the quantum query lower bounds in Section 5, including Theorem 7 and the associated lemmas.

*Proof of Lemma 1.* For the search problem with $m$ copies, we can define the corresponding $m$-player normal-form game with utilities in Definition 5. Then we invoke $\mathcal{A}$ to obtain a set of strategies. For each player's strategy (may be a mixed strategy), we perform a sampling and use the sampled result as the output of the search problem for corresponding copy. The probability that all $m$ copies of the search problem succeed is larger than:

$$(1 - \delta) \cdot \left(1 - \frac{\varepsilon}{B}\right)^m \geq 1 - \left(\delta + \frac{\varepsilon m}{B}\right).$$

Here $(1-\delta)$ is the success probability of algorithm $\mathcal{A}$, and $(1-\frac{\varepsilon}{B})$ is smaller than the probability that one sampling result for index $i$ is exactly corresponded to $k_i$, according to the form of $\varepsilon$-correlated equilibrium in this hard instance. $\qquad\square$

*Proof of Theorem 7.* By Lemma 1, we only need to consider the query lower bound of solving $m$ copies of the search problem. For a single search problem ($m = 1$), it requires $\Omega(\sqrt{n})$ queries to $\mathcal{O}_u$ by quantum query lower bound on unstructured search by Bennett et al. [7]. For general $m$, we leverage the strong direct product theorem provided in Lee and Roland [24], giving a quantum query lower bound on an $m$-copies problem, which shows that computing $m$ copies of a function $f$ needs nearly $m$ times the queries needed for one copy.

**Lemma 4** (Theorem 1.1 in Lee and Roland [24], strong direct product theorem)**.** *Let $f \colon \mathcal{D} \to E$ where $\mathcal{D} \subseteq D^n$ for finite sets $D, E$. For an integer $m > 0$, define $f^{(m)}(x^1, \ldots, x^m) = (f(x^1), \ldots, f(x^m))$. Then, for any $2/3 \leq k \leq 1$,*

$$Q_{1-k^{m/2}}(f^{(m)}) \geq \frac{m \ln(3k/2)}{8000} \cdot Q_{1/4}(f) \ .$$

*Here $Q_\varepsilon(f)$ denotes the query complexity of generating $f$ with error $\varepsilon$.*

Denote $f$ as the search problem of finding $k_i$ for a single $i \in [m]$. Thus we have

$$Q_{1/4}(f) = \Omega(\sqrt{n}).$$

From the above analysis, finding an $\varepsilon$-correlation equilibrium is equivalent to calculating $f^{(m)}$.

Taking $k = (\delta + \frac{\varepsilon}{B}m)^{2/m}$, we have

$$Q_{1-(\delta+\frac{\varepsilon m}{B})}(f^{(m)}) \geq \frac{m \ln(\frac{3}{2} \cdot (\delta + \frac{\varepsilon}{B}m)^{2/m})}{8000} \cdot Q_{1/4}(f)$$

$$= \frac{1}{8000} \cdot \left(m \ln\left(\frac{3}{2}\right) - 2 \ln \frac{1}{\delta + \frac{\varepsilon m}{B}}\right) \cdot Q_{1/4}(f).$$

Taking $\delta = \frac{1}{3}$, the above analysis gives a quantum query lower bound for finding $\varepsilon$-correlated equilibrium with success probability more than $\frac{2}{3}$:

$$\frac{1}{8000} \cdot \left(m \ln\left(\frac{3}{2}\right) - 2 \ln \frac{1}{\delta + \frac{\varepsilon m}{B}}\right) \cdot Q_{1/4}(f) \geq \frac{1}{8000} \cdot \left(m \ln\left(\frac{3}{2}\right) - 2 \ln \frac{1}{\frac{1}{3} + \frac{\varepsilon m}{B}}\right) \cdot Q_{1/4}(f)$$

$$\geq \frac{1}{8000} \cdot \left(m \ln\left(\frac{3}{2}\right) - 2 \ln 3\right) \cdot Q_{1/4}(f)$$

$$= \Omega(m \cdot \sqrt{n}).$$

$\qquad\square$

# D   Impact of quantum sampling noise on the analysis of optimistic MWU

The primary difficulty in extending the proof of Daskalakis et al. [14] to a quantum optimistic MWU algorithm is that the smoothness conditions on the higher-order discrete differentials of the loss vector sequence are violated by the sampling error induced by the quantum Gibbs sampler.

Specifically, let $(D_h \ell)^{(t)} = \sum_{s=0}^{h} \binom{h}{s} (-1)^{h-s} \ell^{(t+s)}$ be the order-$h$ finite difference of the loss vectors $\ell^{(1)}, \ldots, \ell^{(T)}$, as defined in Daskalakis et al. [14, Definition 4.1]. Let $H = \log T$ and $\alpha \in (0, 1/(H+3))$ be two parameters. In a classical $m$-player general-sum game where all players follow OMWU updates with step size $\eta \leq \alpha/(36e^5 m)$, the order-$h$ finite difference of the loss vectors for any player $i$ is bounded by:

$$\|(D_h \ell_i)^{(t)}\|_\infty \leq \alpha^h \cdot h^{3h+1} \tag{62}$$

for all integers $h \in [0, H]$ and $t \in [T - h]$ [14, Lemma 4.4]. This bound is crucial for their main result.

To illustrate the difficulty of extending this proof to a quantum setting, consider a two-player game ($m = 2$). Let $x_i^{(t)}$ be the strategy of player $i \in \{1, 2\}$ at time $t$. In the classical setting, the loss vectors are given by $\ell_1^{(t)} = A_1 x_2^{(t)}$ and $\ell_2^{(t)} = A_2^T x_1^{(t)}$. The proof of Eq. (62) proceeds by induction, first bounding $\|(D_h x_2)^{(t)}\|_1$ via the induction hypothesis and then bounding $\|(D_h \ell_1)^{(t)}\|_\infty$ using the matrix norm inequality:

$$\|(D_h \ell_1)^{(t)}\|_\infty = \left\| A_1 \sum_{s=0}^{h} \binom{h}{s} (-1)^{h-s} x_2^{(t+s)} \right\|_\infty \leq \left\| \sum_{s=0}^{h} \binom{h}{s} (-1)^{h-s} x_2^{(t+s)} \right\|_1 = \|(D_h x_2)^{(t)}\|_1. \tag{63}$$

However, in the quantum setting, we approximate the loss vector $\ell_i^{(t)}$ using a quantum Gibbs sampler with accuracy $\varepsilon_G$, which requires $O(\sqrt{n}/\varepsilon_G^2)$ queries. This introduces an error term. Since $\sum_{s=0}^{h} |\binom{h}{s}| = 2^h$, the inequality in Eq. (63) is weakened to:

$$\|(D_h \ell_1)^{(t)}\|_\infty \leq \|(D_h x_2)^{(t)}\|_1 + 2^h \varepsilon_G. \tag{64}$$

For the original induction scheme to hold, the error term must be absorbed into the bound from Eq. (62). This requires the sampling accuracy $\varepsilon_G$ to satisfy:

$$2^h \varepsilon_G \leq \frac{1}{2} \alpha^h h^{3h+1}. \tag{65}$$

Daskalakis et al. [14] ultimately apply their theorem with $\alpha = 1/(4\sqrt{2}H^{7/2})$. To satisfy Eq. (65), we must therefore choose an $\varepsilon_G$ such that:

$$\varepsilon_G \leq \min_{h \in [0, H]} \frac{1}{2} \cdot 2^{-h} \alpha^h h^{3h+1} \approx \min_{h \in [0, H]} \frac{1}{2} \left( \frac{\alpha h^3}{2} \right)^h = \min_{h \in [0, H]} \frac{1}{2} \left( \frac{h^3}{8\sqrt{2}H^{7/2}} \right)^h.$$

The function $f(h) := \left( \frac{h^3}{8\sqrt{2}H^{7/2}} \right)^h$ attains its minimum at $h = e^{-1}(8\sqrt{2}H^{7/2})^{1/3}$. At this point, the minimum value is approximately:

$$\exp\left( -\frac{3}{e}(8\sqrt{2}H^{7/2})^{1/3} \right) = \exp\left( -\frac{6}{e} 2^{1/6} H^{7/6} \right).$$

Substituting $H = \log(T)$, the required precision becomes $\varepsilon_G = \exp(-\Theta((\log T)^{7/6}))$, which is $o(1/\text{poly}(T))$ for any polynomial in $T$. Consequently, the query complexity of the quantum Gibbs sampler, which scales with $1/\varepsilon_G^2$, becomes superpolynomial in $T$. Since computing an $\varepsilon$-CCE requires setting $T = \tilde{O}(m/\varepsilon)$, this superpolynomial overhead in $T$ translates to a superpolynomial overhead in $m$ and $1/\varepsilon$, rendering the quantum approach impractical under this proof strategy.

