# OpenReview forum: "Near-Optimal Quantum Algorithms for Computing (Coarse) Correlated Equilibria of General-Sum Games"
_NeurIPS.cc/2025/Conference — NeurIPS 2025 poster_

### Official Review · Reviewer_SEAg · 2025-06-22

**Clarity:** 3
**Significance:** 2
**Originality:** 2
**Rating:** 4
**Confidence:** 3

**Summary:**

This paper studies the theoretical convergence rates of quantum algorithms for computing correlated and coarse correlated equilibria in normal form general-sum games. The key results in this paper pertain to the introduction of a multi-scale MWU algorithm for computing CE, and a sample-based MWU algorithm for computing CCE. The analysis of these algorithm results in rates which are near-optimal (compared to the corresponding classical variants of the algorithms) in terms of scaling in the number of players and actions of the game. Meanwhile, the paper also gives lower bounds on the query complexity of computing both approximate CE and CCE in general-sum games.

**Questions:**

- Can you clarify on the roadblocks towards quantizing the Optimistic MWU algorithm? The difficulties are mentioned in passing in the main text but I would like to understand the details on how the smoothness assumption of the loss vectors would lead to the analysis failing.

**Ethical Concerns:**

["NO or VERY MINOR ethics concerns only"]

**Final Justification:**

- My concerns about the quantization of the Grigoriadis and Khachiyan algorithm as opposed to other 'near-optimal' algorithms have been alleviated.
- A remaining concern regards the structure of the paper, and making it more readable for a neurips audience. However, given the authors suggestions (making space for more quantum algorithm preliminaries), I think this can be done in a camera ready version should the paper be accepted.
- I will update my score to a borderline accept -- while the authors have addressed my major concern, I still feel that the paper needs a bit of work to be readable to a general audience. Moreover, having more concrete discussion/analysis towards quantizing the optimistic algorithms which are considered 'standard' in the classical literature would make the paper a lot stronger.

**Limitations:**

Yes.

**Paper Formatting Concerns:**

No major formatting concerns. Some minor typos:
- thoery on line 4
- MUW_k on line 240

**Quality:**

3

**Strengths And Weaknesses:**

Strengths: The paper is fairly well written and has clear exposition. The motivation to initiate the theoretical complexity analysis of CE and CCE computation using quantum algorithms is also strong and relevant to the current framework of classical learning in games. The multiscale MWU algorithm for computing CE extends recent work by Peng and Rubinstein/Dagan et al to a quantum setting.

Weaknesses: While the theoretical rates are near optimal as compared to existing classical algorithms, I would have appreciated more thorough comparison between the classical and quantum time complexity, and when the quantum algorithms exhibit speedup (for example, in terms of time complexity the quadratic speedup in n comes at the cost of worse dependence on m). To my knowledge, much of the technical analysis is derived similarly to prior works, and if there is significant technical novelty here it is not made clear. It is also unclear to me why the authors focus on the algorithm of Grigoriadis and Khachiyan for CCE computation, when there are several recently proposed MWU variants that have better query/time complexity. For example, in addition to Optimistic MWU, some recent papers have introduced Clairvoyant [1] and Cautious [2] MWU which have significantly improved rates for CCE computation in general-sum games. The authors discuss the difficulties of replicating the proof of Daskalakis et al for OMWU, but fail to discuss the original RVU bound in [3]. Indeed, the proof of Daskalakis et al is highly intricate and I do not doubt that quantizing the analysis would be challenging, but the original bound in [3] would be more feasible as a first step.

[1] Piliouras, G., Sim, R., & Skoulakis, S. (2022). Beyond time-average convergence: Near-optimal uncoupled online learning via clairvoyant multiplicative weights update. Advances in Neural Information Processing Systems, 35, 22258-22269.

[2] Soleymani, A., Piliouras, G., & Farina, G. (2025). Cautious Optimism: A Meta-Algorithm for Near-Constant Regret in General Games. arXiv preprint arXiv:2506.05005.

[3] Syrgkanis, V., Agarwal, A., Luo, H., & Schapire, R. E. (2015). Fast convergence of regularized learning in games. Advances in Neural Information Processing Systems, 28.

Overall, while the results in this paper are a good first step in terms of CE and CCE computations with quantum algorithms, I am unconvinced as to the relevance/usefulness of the CCE algorithm introduced as compared to modern (classical) algorithms such as optimistic MWU.

---

> ### Author Rebuttal · Authors · 2025-07-30
>
> We thank the reviewer for their valuable feedback. We address the specific points raised in the review below.
>
> **Detailed response to Weaknesses:**
>
> We thank the reviewer for their insightful comments and constructive feedback, which will help us improve the manuscript.
>
> Our manuscript focuses on the complexity of computing CE and CCE as a function of the number of players $m$ and the number of actions $n$. While in self-play settings, each player running Optimistic MWU [3] or Clairvoyant and Cautious MWU [1,2] achieves $o(T^{1/2})$ regret, the $m$ dependence is $O(\sqrt{m})$ or $O(m)$. Since we still need $O(m)$ queries per iteration to update the strategies of all $m$ players, the total query complexity of the CCE computation algorithm is super-linear in $m$, which is suboptimal. In contrast, the Grigoriadis and Khachiyan algorithm achieves a regret bound that is independent of the number of players. This property allows us to compute CCE with optimal query complexity in $m$. We will make sure to clarify this important motivation in the revised manuscript.
>
> Regarding the point on improving the error dependence, we agree with the reviewer that extending the RVU bound in [3] to quantum cases is a good start. Its reliance on first-order discrete differentials of the loss vector sequence, rather than the higher-order ones in Daskalakis et al., makes the analysis more robust to the sample error induced by the quantum Gibbs sampler. Furthermore, the Cautious MWU algorithm [2], which builds on a non-negative RVU bound and was posted online after our manuscript's submission at NeurIPS 2025, would be a next step in this direction. This is an excellent suggestion, and we will add a discussion of this promising future direction to our paper.
>
> **Detailed response to Questions:**
>
>
>
> The primary difficulty in extending the proof of Daskalakis et al. to a quantum Optimistic MWU algorithm is that the smoothness conditions on the higher-order discrete differentials of the loss vector sequence are violated by the sampling error induced by the quantum Gibbs sampler.
>
> Specifically, let $(  \mathrm{D}\_h\ell)^{(t)}=\sum\_{s=0}^h\binom{h}{s}(-1)^{h-s} \ell^{(t+s)}$ be the order-$h$ finite difference of the loss vectors $\ell^{(1)}, \dots, \ell^{(T)}$, as defined in [Definition 4.1, Daskalakis et al.]. Let $H = \log T$ and $\alpha\in (0, 1/(H+3))$ be two parameters. In a classical $m$-player general-sum game where all players follow OMWU updates with step size $\eta\le \alpha/(36e^5m)$, the order-$h$ finite difference of the loss vectors for any player $i$ is bounded by:
> \begin{equation}
> \tag{1}
>     \|(  \mathrm{D}\_h \ell\_i)^{(t)}\|\_{\infty} \leq \alpha^h \cdot h^{3 h+1}
> \end{equation}
> for all integers $h\in [0, H]$ and $t\in [T-h]$ [Lemma 4.4, Daskalakis et al.]. This bound is crucial for their main result.
>
> To illustrate the difficulty of extending this proof to a quantum setting, consider a two-player game ($m=2$). Let $x\_i^{(t)}$ be the strategy of player $i\in \{1,2\}$ at time $t$. In the classical setting, the loss vectors are given by $\ell\_1^{(t)}=A\_1 x\_2^{(t)}$ and $\ell\_2^{(t)}=A\_2^{\top} x\_1^{(t)}$. The proof of Eq. (1) proceeds by induction, first bounding $\|(  \mathrm{D}\_h x\_2)^{(t)}\|\_1$ via the induction hypothesis and then bounding $\|(  \mathrm{D}\_h \ell\_1)^{(t)}\|\_{\infty}$ using the matrix norm inequality:
> \begin{equation}
> \tag{2}
>     \|(  \mathrm{D}\_h \ell\_1)^{(t)}\|\_{\infty}=\left\|A\_1 \sum\_{s=0}^h\binom{h}{s}(-1)^{h-s} x\_2^{(t+s)}\right\|\_{\infty} \leq \left\|\sum\_{s=0}^h\binom{h}{s}(-1)^{h-s} x\_2^{(t+s)}\right\|\_1=\|(  \mathrm{D}\_h x\_2)^{(t)}\|\_1.
> \end{equation}
> However, in the quantum setting, we approximate the loss vector $\ell\_i^{(t)}$ using a quantum Gibbs sampler with accuracy $\varepsilon\_G$, which requires $O(\sqrt{n}/\varepsilon\_G^2)$ queries. This introduces an error term. Since $\sum\_{s=0}^h |\binom{h}{s}| = 2^h$, the inequality in Eq. (2) is weakened to:
> \begin{equation}
>     \|(  \mathrm{D}\_h \ell\_1)^{(t)}\|\_{\infty}\le\|(  \mathrm{D}\_h x\_2)^{(t)}\|\_1+2^h \varepsilon\_G.
> \end{equation}
> For the original induction scheme to hold, the error term must be absorbed into the bound from Eq. (1). This requires the sampling accuracy $\varepsilon\_G$ to satisfy:
> \begin{equation}
> \tag{3}
>     2^h \varepsilon\_G\le \frac{1}{2} \alpha^h h^{3h+1}.
> \end{equation}
> Daskalakis et al. ultimately apply their theorem with $\alpha = 1/(4\sqrt{2}H^{7/2})$. To satisfy Eq. (3), we must therefore choose an $\varepsilon\_G$ such that:
> $$
> \varepsilon\_G \le \min\_{h\in [0,H]} \frac{1}{2} \cdot 2^{-h} \alpha^h h^{3h+1} \approx \min\_{h\in [0,H]} \frac{1}{2}\left(\frac{\alpha h^3}{2}\right)^h = \min\_{h\in [0,H]} \frac{1}{2}\left(\frac{h^3}{8\sqrt{2}H^{7/2}}\right)^h.
> $$
> The function $f(h):=\left(\frac{ h^3}{8\sqrt{2}H^{7/2}}\right)^h$ attains its minimum at $h=e^{-1}(8\sqrt{2}H^{7/2})^{1/3}$. At this point, the minimum value is approximately:
> $$
> \exp\left(-\frac{3}{e}(8\sqrt{2}H^{7/2})^{1/3}\right) = \exp\left(-\frac{6}{e}2^{1/6} H^{7/6}\right).
> $$
> Substituting $H=\log(T)$, the required precision becomes $\varepsilon\_G = \exp(-\Theta((\log T)^{7/6}))$, which is $o(1/\mathrm{poly}(T))$ for any polynomial in $T$. Consequently, the query complexity of the quantum Gibbs sampler, which scales with $1/\varepsilon\_G^2$, becomes superpolynomial in $T$. Since computing an $\varepsilon$-CCE requires setting $T = \tilde{O}(m/\varepsilon)$, this superpolynomial overhead in $T$ translates to a superpolynomial overhead in $m$ and $1/\varepsilon$, rendering the quantum approach impractical under this proof strategy.
>
> **Response to Paper Formatting Concerns**
>
> Thank you for identifying these typos. We will fix them in the revised manuscript.

---

> > ### Comment · Reviewer_SEAg · 2025-08-01
> > **Response to Author Rebuttal**
> >
> > Thank you for your response. I appreciate the detailed discussion about the sampling error introduced by the quantum Gibbs sampler, it has clarified my question quite thoroughly. I also understand now the need to reduce dependence on number of players in order to reduce the QRAM size required. Overall, I am happy to increase my score to a marginal accept, especially if the authors are also intending to add more clarifying exposition and clean the main text to be more suitable for the Neurips audience as suggested by Rev. 42NM.

---

> > > ### Author Response · Authors · 2025-08-01
> > >
> > > Thank you for your positive feedback and for increasing your score! We're glad our discussion on sampling error resolved your question.
> > >
> > > We fully agree with you and Reviewer 42NM on the need to make the paper more suitable for the NeurIPS audience.
> > >
> > > As suggested, we will move the algorithms for prior work (Algs. 1 & 2) to the appendix. Then we will provide a more thorough introduction to foundational concepts like the quantum query model and QRAM, making the paper more self-contained and accessible. We will also add a more detailed discussion of how our algorithm achieves a linear dependence on the number of players $m$.

---

### Official Review · Reviewer_y8aw · 2025-06-28

**Clarity:** 3
**Significance:** 3
**Originality:** 3
**Rating:** 4
**Confidence:** 2

**Summary:**

This paper introduces quantum algorithms for computing correlated equilibria (CE) and coarse correlated equilibria (CCE) in multi-player general-sum normal-form games. Building on quantum improvements to the multiplicative weights update (MWU) method, the authors design algorithms that achieve query complexities of $\tilde{O}(m\sqrt n)$ for CE and $\tilde{O}(m\sqrt n / \epsilon^{2.5})$ for CCE.
The algorithms are proven to be nearly optimal by establishing matching lower bounds of $\Omega(m\sqrt n)$.

**Questions:**

To what extent do the proposed algorithms require genuinely new quantum techniques, as opposed to adapting known primitives (e.g., quantum Gibbs sampling and amplitude encoding) within existing classical frameworks like MWU? Are there aspects of the analysis or algorithm design that are specific to the quantum regime?

**Ethical Concerns:**

["NO or VERY MINOR ethics concerns only"]

**Final Justification:**

The authors have addressed my questions. I will retain my score.

**Limitations:**

No limitations

**Paper Formatting Concerns:**

No concerns

**Quality:**

3

**Strengths And Weaknesses:**

* The paper makes an interesting contribution by extending quantum algorithmic techniques, previously developed mainly for zero-sum games, to the broader and more challenging setting of general-sum games
* The authors establish matching lower bounds that confirm the near-optimality of their algorithms, giving strong theoretical support to the efficiency of their proposed methods.
* The paper is well written, mathematically precise, and the results are well situated within the broader literature on quantum optimization and equilibrium computation.

---

> ### Author Rebuttal · Authors · 2025-07-30
>
> We thank the reviewer for their valuable feedback. We address the specific points raised in the review below.
>
> **Detailed response to Questions:**
>
> Thank you for this question. The new quantum technique in our work is the method for constructing the amplitude encoding. Instead of following prior methods that build a frequency vector of actions, which would require a QRAM of size $\Omega(n^m)$, our algorithm stores the history of actions and builds amplitude encoding from them (see Section A.1 and A.3). This approach significantly reduces the QRAM size, which is a critical bottleneck for many quantum algorithms.
>
> The analysis of our algorithm has aspects that are both quantum-specific and shared with classical analysis. The analysis of the amplitude encoding construction, its gate complexity, and how it is used to build the amplitude encoding for the loss vectors are all inherently quantum. The analysis of how sampling errors from the Gibbs sampler affect the algorithm's performance has common ground with classical online learning. This part of the analysis addresses whether an online learning algorithm can still guarantee low regret when it receives a noisy loss vector (see Section A.2 and A.4).

---

### Official Review · Reviewer_42NM · 2025-07-02

**Clarity:** 2
**Significance:** 3
**Originality:** 3
**Rating:** 4
**Confidence:** 3

**Summary:**

This paper initiates a study on quantum algorithms for computing ε-approximate Correlated Equilibria (CE) and Coarse Correlated Equilibria (CCE) in multi-player normal-form games.
Classical computation of Nash equilibria in general-sum games is well explored but highly intractable. Quantum algorithms can potentially accelerate the computation, but haven’t been studied much in normal form games > 2 players.

**Questions:**

1. How is the failure probability handled $\alpha$ in the complexity analysis? Are the reported query and time complexities assuming a fixed failure probability? In particular, the lower bound results (e.g., Theorem 7 and Corollary 2) assume a fixed success probability of 2/3. Can I still compare those lower bounds directly with the upper bounds in Tables 2 and 3?

2. I have an intuition-related question: in Bouland et al. (2023), the query complexity for computing an approximate Nash equilibrium in two-player zero-sum games is $\widetilde{O}(\sqrt{2n} / \varepsilon^{2.5})$. In your Table 3, the query complexity for computing an ε-CCE is $\widetilde{O}(2 \sqrt{n} / \varepsilon^{2.5})$ for 2-player. While the difference is just a constant, does this suggest a broader generalization to $\widetilde{O}(\sqrt{mn} / \varepsilon^{2.5})$?

3. Also, Bouland’s work has a $\varepsilon^{-3}$ term. Could you clarify why this term does not appear in the complexity of your CCE algorithm?

**Ethical Concerns:**

["NO or VERY MINOR ethics concerns only"]

**Final Justification:**

The authors’ reply clarified my concerns regarding the methodology and novelty. However, the current presentation of the paper does not fully meet the expected NeurIPS standard. Due to the conference policy, which restricts revisions to the PDF and enforces a limited rebuttal period, the evaluation is largely based on the submitted version. The score will therefore remain unchanged.

**Limitations:**

yes

**Quality:**

3

**Strengths And Weaknesses:**

**Strength**

- The first quantum algorithms for computing approximate CE and CCE, which are important and non-trivial extensions of existing work on multi-player games + quantum algorithms.

- The proposed algorithms achieve quadratic speedups in terms of the number of actions n, compared to known classical algorithms.

- The paper provides detailed proofs and complexity analyses. While I cannot fully verify the correctness of all proofs, the results appear to be presented in a reasonable and well-structured manner.

**Weakness**

- There are relatively few quantum algorithm papers at Neurips, and I found the structure of this one difficult to follow for outsiders. There are too many technical details before introducing the necessary preliminaries, which makes it challenging to understand the main ideas early on. The paper "Logarithmic-Regret Quantum Learning Algorithms for Zero-Sum Games" could be a good reference.

- Algorithms 1 and 2 are existing methods from prior work, and thus could be moved to the appendix. This would free up space for a proper conclusion or wrap-up discussion at the end of the main text.

---

> ### Author Rebuttal · Authors · 2025-07-30
>
> We thank the reviewer for their valuable feedback. We address the specific points raised in the review below.
>
> **Detailed response to Weaknesses:**
>
> We thank the reviewer for the constructive feedback on the paper's presentation.
>
> (1) We agree that the main ideas should be more accessible early on. We will revise the introduction to better motivate our work and improve the overall flow.
>
> (2) This is a good suggestion. We will move Algorithms 1 and 2 to the appendix and introduce the prior work more concisely in the main text.
>
> **Detailed response to Questions:**
>
> We sincerely thank the reviewer for their insightful questions, which help us refine our manuscript.
>
> (1) The reported query and time complexities in our work are not for a fixed failure probability.
> For the sake of brevity, the query and time complexities stated in our main theorems (Theorem 1 and Theorem 2) and Table 2 and 3 use the $\tilde{O}$ notation, which hides the logarithmic factors in $\frac{1}{\alpha}$. For CE, the dependence of $\alpha$ is shown in Theorem 8; for CCE, the dependence of $\alpha$ can be found in Lemma 3. (We will clarify this in Theorem 9 of our revised manuscript.) This logarithmic dependence arises from a standard amplification technique allowing us to convert an algorithm with a fixed success probability into one with an arbitrarily high success probability by running the algorithm independently multiple times, and the final failure probability decreases exponentially with the number of repetitions. This boosting strategy works for both classical algorithms and quantum algorithms, and it is applied in the subroutine in our CE and CCE algorithms.
>
> Our quantum query lower bounds (Theorem 7 for CE and Corollary 2 for CCE) are proven with a fixed success probability of 2/3, which is a common and standard practice in quantum query complexity. The $\Omega(m\sqrt{n})$ term represents the complexity for the problem dimension, and is consistent with the upper bounds in Tables 2
> and 3.
>
> (2) It should be noted that both our work and Bouland et al. (2023) use $\tilde{O}$ notation, which hides constant factors of the complexity. For the case of $m=2$, our algorithm matches the complexity of Bouland et al. (2023), achieving a $\sqrt{n}$ dependence. For a general $m$-player setting, with $m$ held constant, our algorithm retains this $\sqrt{n}$ complexity. When considering arbitrary $m$, the complexity scales as $m \sqrt{n}$. Notably, our lower bound is also $m \sqrt{n}$, which implies that an algorithm with complexity $\sqrt{mn}$ do not exist.
>
> (3) This is a good observation.
> First, there is a subtle difference between Bouland et al.'s work and ours regarding the assumptions on QRAM. Their QRAM model assumes that mathematical operations can be implemented exactly in $O(1)$ time. In contrast, we further consider the gate complexity of QRAM operations in our analysis. Their $\epsilon^{-3}$ term actually arises from an additive initialization cost $\tilde{O}(\eta^3T^3)$ in Lemma 2, which is unrelated to the number of queries to the loss oracle $O_\mathcal{L}$ and appears only in the time complexity. When considering query complexity, Bouland et al.’s dependence on $\epsilon$ is $\tilde{O}({1/\epsilon^{2.5}})$, which matches ours exactly. However, for the time complexity, due to our additional consideration of the gate complexity of QRAM operations, our overall time complexity becomes $\tilde{O}({1/\epsilon^{4.5}})$, which is larger than $\epsilon^{-3}$. Therefore, we do not explicitly include the additive initialization cost term $\epsilon^{-3}$ in the final stated result.

---

> > ### Comment · Reviewer_42NM · 2025-08-05
> >
> > Thank you for the detailed reply. I appreciate the clarifications and the additional insights provided. I am currently debating between an accept and a borderline accept.
> > I must admit I’m not a fan of the current conference policy, which enforces a tight rebuttal schedule and does not allow any updates to the PDF. Unfortunately, given this restriction, I will keep my score unchanged based on the current version of the paper.

---

> > > ### Author Response · Authors · 2025-08-05
> > >
> > > Thank you for the positive feedback on our rebuttal and for your support. Please let us know if you have any further questions or if any additional clarifications would be helpful.

---

### Official Review · Reviewer_4Fe3 · 2025-07-04

**Clarity:** 2
**Significance:** 2
**Originality:** 2
**Rating:** 4
**Confidence:** 4

**Summary:**

The paper proposes quantum algorithms for computing $\epsilon$-approximate CE and CCE in multi-player normal-form games. The authors also give quantum lower bounds on the query complexity for both CE and CCE, showing that both proposed quantum algorithms achieve nearly-optimal query complexity in the number of players and actions.

**Questions:**

(1) In Def 1, the paper gives the description of the loss oracle $\mathcal{O}_{\mathcal{L}}$. Is it efficient to build this oracle from the corresponding classical oracle? I think it depends on the distributions over the action sets, does it?
(2) CE is more general than Nash equilibrium. But does it make CE more difficult than Nash equilibrium, complexity-wise?
(3) related to the weaknesses mentioned above.

**Ethical Concerns:**

["NO or VERY MINOR ethics concerns only"]

**Final Justification:**

I recommend to accept this paper because this work proposes quantum algorithms and proves a quantum lower bound as well.

**Limitations:**

Yes

**Quality:**

3

**Strengths And Weaknesses:**

Overall, the paper is well-organized and well-written. I didn't check all the proofs in detailed, but they seem correct.

Strengths:
(1) The paper initiate the study of quantum algorithms for $\epsilon$-approximate CE and CCE problems in multi-player normal-form games.
(2) The paper also proves the quantum lower bounds on the query complexity of both problems by reducing the direct product of $m$ instances of the unstructured search problem to the  $\epsilon$-approximate CE and CCE problems in multi-player normal-form games.

Weaknesses:
(1) I couldn't find the details of how to construct the QRAM for the history action samples. I think it will be necessary have those details presented with the algorithms, because the QRAM is needed for the amplitude-encoding of loss vectors which is very important for the quantum algorithms to achieve better performance.
(2) Most of time, QRAM is powerful and too good to have. This also sort of means that the construction of a QRAM useful for solving the problem is often not efficient enough. As the time complexity shown in table 2 and 3, it has $O(1/\epsilon)$ in the exponent, which will bring the time to exponential if wanting $1/poly(n,m)$ approximation ratio. Hence, I think it will enrich the paper by providing some discussion on this issue.

Minor issues on some clarification of notations:
(1) formal definition of $\delta(A)$?
(2) in line 208, does $q$ represent the vector $(q_1, \cdots,  q_n)$?
(3) in Def 4, $|\psi_i \rangle$ in reg B can be regarded as garbage?

---

> ### Author Rebuttal · Authors · 2025-07-30
>
> We thank the reviewer for their valuable feedback. We address the specific points raised in the review below.
>
> **Detailed response to Weaknesses:**
>
> We thank the reviewer for raising this important question. We note that the term "QRAM" can refer to different concepts in the literature (see the survey "QRAM: A Survey and Critique" by Jaques and Rattew). In our work, we use it in the sense of a circuit for accessing classical data---often specifically called a Quantum Read-Only Memory (QROM)---which is significantly less complicated to construct than a QRAM designed to store quantum states. We agree that the details of this construction are essential, and we will add a more detailed explanation to the manuscript based on the following arguments:
>
> (1) In our proposed algorithm, this QROM stores classical data. Specifically, after obtaining the classical descriptions of the history actions $a^{(\tau,s)}$ (as in Line 6 of Algorithm 3), we store them in a classical table. The algorithm runs for $T$ rounds and generates $S$ samples per iteration, resulting in a total of $TS$ actions that need to be stored. Since each action requires $m\log(n)$ bits for its classical description, the total size of this table is $TSm\log(n)$ bits.
>
> The function of the QROM is to make this classical data accessible to the quantum computer via a unitary operation, $U_{\mathrm{QRAM}}$, that performs the mapping:
> $$
> U_{\mathrm{QRAM}}\colon \ket{\tau}\ket{s}\ket{0} \mapsto \ket{\tau}\ket{s}\ket{a^{(\tau,s)}}.
> $$
> Given the classical table, a standard construction for $U_{\mathrm{QRAM}}$ proceeds as follows: for all pairs $(\tau, s)$ where $\tau \in [T]$ and $s \in [S]$, one implements a controlled operation that checks if the address register is in the state $\ket{\tau}\ket{s}$. If the condition is met, the corresponding classical data $a^{(\tau,s)}$ is written to the data register. These controlled operations can be decomposed into a sequence of Toffoli and X gates. The gate complexity of this construction is $O(TSm\log(TSmn))$  and can be further optimized to $O(TSm\log(n))$ using the QROM construction from "Encoding Electronic Spectra in Quantum Circuits with Linear T Complexity" by Babbush et al. In our complexity analysis, for instance in the paragraph before Eq.~(34), we use this optimal $O(TSm\log(n))$ gate complexity.
>
> (2) We agree that QRAM can be a resource-intensive component, and this is precisely why our analysis does **not** assume access to a free QRAM oracle. Instead, our time complexity analysis explicitly accounts for the number of elementary gates required to construct the QRAM.
> This is the reason why our overall time complexity has an $O(m)$ overhead compared to the query complexity. As we mention in the "Techniques" paragraph, this $O(m)$ overhead is a significant achievement of our work, as it reduces the $O(n^m)$ gate cost required to construct the QRAM that arises from a direct extension of prior quantum algorithms for two-player games to the $m$-player setting.
>
> We also wish to clarify that the $O(1/\epsilon)$ term in the exponent is **not** an artifact of the QRAM construction overhead. This dependence is inherent to the swap-regret-minimization problems.
> For instance, the classical Correlated Equilibrium (CE) algorithms of Peng and Rubinstein [25] and Dagan et al. [11] also exhibit this exponential dependence on $1/\epsilon$. Their work further establishes a fundamental lower bound: achieving an $\epsilon$-swap regret requires a number of rounds that has either a polynomial dependence on $n$ (i.e., $\tilde{\Omega}(n/\epsilon^2)$) or is exponential in $1/\epsilon$.
>
> Minor issues: (1) $\Delta(\mathcal{A})$ is the class of probability distributions on $\mathcal{A}$.
> (2) Yes.
> (3) Yes.  We thank the reviewer for pointing out these minor issues and for the helpful suggestions. We will clarify these points in the revised manuscript.
>
> **Detailed response to Questions:**
>
> Thank you for these insightful questions. We will add these clarifications to the manuscript.
>
> (1) This is a good question. The quantum loss oracle $\mathcal{O}_{\mathcal{L}}$ can indeed be constructed efficiently if the corresponding classical oracle is efficient. It is a standard result in computational complexity that any classical circuit that computes a function in time $T$ and space $S$ can be converted into a reversible quantum circuit with a minimal increase in resources (e.g., time $T^{1+\epsilon}$ and space $S\log T$ for any $\epsilon > 0$, as shown in "Time/Space Trade-Offs for Reversible Computation" by Bennett). The efficiency of this construction depends on the circuit complexity of the classical oracle itself, not on the input distributions over the action sets.
>
> (2) You are correct that Correlated Equilibrium (CE) is a more general solution concept than Nash Equilibrium (NE). However, for the computational task of finding an equilibrium, this generality makes the problem easier. Since every Nash Equilibrium is also a Correlated Equilibrium, the set of CEs is a superset of the set of NEs. This larger solution space makes the search problem of finding a CE computationally easier than finding an NE in general-sum games.

---

> > ### Comment · Reviewer_4Fe3 · 2025-08-05
> > **Response to Authors' Rebuttal**
> >
> > Thanks for the detailed reponse. I have this one last question: I understand that an efficient classical construction will lead to an efficient quantum one immediately by standard techniques. But more specifically in your case, what is your classical oracle like? Can you give a concrete example? I think this will help me better understand the information provided by the oracle.

---

> > > ### Author Response · Authors · 2025-08-05
> > >
> > > Thank you for the question. To illustrate how the loss oracle is implemented, we use the congestion game as a concrete example.
> > >
> > > Consider a setting with $m$ players and $n$ resources. The core mechanic is that when multiple players use the same resource, it becomes more costly for every one of its users.
> > >
> > > In this context, the oracle computes a player's loss as follows:
> > >
> > > Given a specific action (i.e., a chosen set of resources) from every player, it first calculates the congestion on each resource by counting how many players chose it. Based on this congestion, it then sums the costs of the resources selected by the particular player in question to get their total loss.

---

> > > > ### Comment · Reviewer_4Fe3 · 2025-08-07
> > > >
> > > > Thanks for providing the example for my last question. I would suggest to add an example like this to the work. This can serve as an evidence that the corresponding quantum oracle can be efficiently constrcuted from its classical counterpart. This can be very helpful because there are problems where classical oracles are natural and formed as black-box and building the quantum oracles from them can probably be not easier than solving the original problems.

---

> > > > > ### Author Response · Authors · 2025-08-07
> > > > >
> > > > > Thank you for the good suggestion. We will incorporate the congestion game example into the revised manuscript.

---

### Note · Authors · 2025-08-12

Dear Area Chair,

We thank the reviewers for their constructive feedback and will incorporate all suggested revisions. Below is a summary of our contributions and our responses to the main reviewer concerns.

### Our Contributions:

Our paper introduces the first quantum algorithms for computing correlated equilibrium and coarse correlated equilibrium in general-sum games. Our key results are:

* **Quadratic speedup in actions ($n$):** Our algorithms have $\tilde{O}(m\sqrt{n})$ query complexity, a quadratic improvement in the dependence on $n$ compared to the $\tilde{O}(mn)$ of standard classical algorithms.

* **Near-optimal dependence on actions ($n$) and players ($m$):** The algorithm's complexity matches our proven $\Omega(m\sqrt{n})$ quantum lower bound.

* **Key Technical Innovation:** We provide a gate-level analysis of a novel amplitude encoding method. This avoids the exponential $\Omega(n^m)$ QRAM cost of previous approaches and does not assume a free QRAM oracle, which is a common bottleneck of quantum algorithms.

### Response to Reviewer Concerns:

* **Clarity:** As requested, we will improve the paper's accessibility by revising the introduction, adding a quantum preliminaries section for the NeurIPS audience, and moving prior work to the appendix.

* **Technical Details:** We clarified that our QRAM analysis has always included gate complexity and explained how our algorithm's $\epsilon$-dependence differs from prior work. To demonstrate the practicality of our loss oracle, we provided a concrete congestion-game example, which we will add to the revised manuscript as suggested by the reviewer.

* **Algorithmic Choice:** We justified our choice of the Grigoriadis-Khachiyan framework because its regret is independent of player count ($m$), which is essential for our results. In contrast, quantizing the recent analysis of Optimistic MWU (OMWU) leads to a suboptimal dependence on $m$.

* **Quantum Barriers:** We provided a detailed mathematical proof showing why that OMWU analysis is difficult to quantize: sampling errors from the required quantum Gibbs sampler violate its core smoothness assumptions, leading to superpolynomial complexity.

The reviewers' feedback has significantly strengthened our paper. We believe the revised manuscript offers a valuable contribution to the fields of online learning, algorithmic game theory, and quantum computation. Thank you for your consideration.

---

### Decision · Program_Chairs · 2025-09-17

**Decision:**

Accept (poster)

**Comment:**

This paper introduces the first quantum algorithm for computing correlated equilibria (CE) and coarse correlated equilibria (CCE) in general-sum games, which are key solution concepts in algorithmic game theory. Its key contributions include a set of algorithms that provide a quadratic speedup in the number of actions compared to classical methods. Complementary to these upper bounds the paper provides new quantum query lower bounds, demonstrating that their algorithms' scaling in the number of players and actions is near-optimal. The work is technically solid and initiates a new direction at the intersection of quantum computation and multi-agent systems. Given other recent work in terms of quantum generalizations of normal-form games and their corresponding solution concepts (e.g. quantum CCEs) (see [1-4] and references therein) I believe that this paper could trigger advancements in other related areas in the intersection of these two areas. I would encourage the authors to examine how the literature above could inform the frontier of open problems in this area. Moreover, the faster learning algorithms for classical computation of CCE are now corresponding to the cautious optimism framework (see [5,6]). Lastly, as requested, make sure to improve the paper's accessibility by revising the introduction and prelims appropriately for the NeurIPS audience.


[1] Jain, Rahul, Georgios Piliouras, and Ryann Sim. "Matrix multiplicative weights updates in quantum zero-sum games: Conservation laws & recurrence." Advances in Neural Information Processing Systems 35 (2022): 4123-4135.
[2] Lotidis, Kyriakos, Panayotis Mertikopoulos, and Nicholas Bambos. "Learning in quantum games." arXiv preprint arXiv:2302.02333 (2023).
[3] Lin, Wayne, et al. "No-Regret Learning and Equilibrium Computation in Quantum Games." Quantum 8 (2024): 1569.
[4] Vasconcelos, Francisca, et al. "A quadratic speedup in finding Nash equilibria of quantum zero-sum games." Quantum 9 (2025): 1737.
[5] Soleymani, Ashkan, Georgios Piliouras, and Gabriele Farina. "Faster Rates for No-Regret Learning in General Games via Cautious Optimism." Proceedings of the 57th Annual ACM Symposium on Theory of Computing. 2025.
[6] Soleymani, Ashkan, Georgios Piliouras, and Gabriele Farina. "Cautious Optimism: A Meta-Algorithm for Near-Constant Regret in General Games." arXiv preprint arXiv:2506.05005 (2025).